# Is Deeper Better only when Shallow is Good?

**Eran Malach**
School of Computer Science
The Hebrew University
Jerusalem, Israel
eran.malach@mail.huji.ac.il

**Shai Shalev-Shwartz**
School of Computer Science
The Hebrew University
Jerusalem, Israel
shais@cs.huji.ac.il

## Abstract

Understanding the power of depth in feed-forward neural networks is an ongoing challenge in the field of deep learning theory. While current works account for the importance of depth for the expressive power of neural-networks, it remains an open question whether these benefits are exploited during a gradient-based optimization process. In this work we explore the relation between expressivity properties of deep networks and the ability to train them efficiently using gradient-based algorithms. We give a depth separation argument for distributions with fractal structure, showing that they can be expressed efficiently by deep networks, but not with shallow ones. These distributions have a natural coarse-to-fine structure, and we show that the balance between the coarse and fine details has a crucial effect on whether the optimization process is likely to succeed. We prove that when the distribution is concentrated on the fine details, gradient-based algorithms are likely to fail. Using this result we prove that, at least in some distributions, the success of learning deep networks depends on whether the distribution can be approximated by shallower networks, and we conjecture that this property holds in general.

## 1 Introduction

A fundamental question in studying deep networks is understanding why and when "deeper is better". In recent years there has been a large number of works studying the expressive power of deep and shallow networks. The main goal of this research direction is to show families of functions or distributions that are realizable with deep networks of modest width, but require exponential number of neurons to approximate by shallow networks. We refer to such results as depth separation results.

Many of these works consider various measures of "complexity" that can grow exponentially fast with the depth of the network, but not with the width. Hence, such measures provide a clear separation between deep and shallow networks. For example, the works of [11, 10, 9, 16] show that the number of linear regions grows exponentially with the depth of the network, but only polynomially with the width. The works of [13, 14, 20] give similar results for other complexity measures, such as the curvature, the trajectory length or the number of oscillations of the output function.

While such works give general characteristics of function families, they take a seemingly worst-case approach. Hence, it is not clear whether such analysis applies to the typical cases encountered in the practice of neural-networks. To answer this concern, recent works show depth separation results for narrower families of functions that appear simple or "natural". For example, the work of [19] shows a very simple construction of a function on the real line that exhibits depth separation. The works of [6, 15] show a depth separation argument for very natural functions, like the indicator function of the unit ball. The work of [3] gives similar results for a richer family of functions. Other works by [8, 12] show that compositional functions, namely functions of functions, are well approximated by deep networks. The works of [4, 2] show similar depth separation results for sum-product networks.

While current works provide a variety of depth separation results, these are all focused on expressivity analysis. This is unsatisfactory, as the fact that a certain network architecture can express some function does not mean that we can learn it from training data in reasonable training time. In fact, there is theoretical evidence showing that gradient-based algorithms can only learn a small fraction of the functions that are expressed by a given neural-network (e.g [18]).

This paper relates expressivity properties of deep networks to the ability to train them efficiently using a gradient-based algorithm. We start by giving depth separation arguments for distributions with fractal structure. In particular, we show that deep networks are able to exploit the self-similarity property of fractal distributions, and thus realize such distributions with a small number of parameters. On the other hand, we show that shallow networks need a number of parameters that grows exponentially with the intrinsic "depth" of the fractal. The advantage of fractal distributions is that they exhibit a clear coarse-to-fine structure. We show that if the distribution is more concentrated on the "coarse" details of the fractal, then even though shallower networks cannot exactly express the underlying distribution, they can still achieve a good approximation. We introduce the notion of **approximation curve**, that characterizes how the examples are distributed between the "coarse" details and the "fine" details of the fractal. The approximation curve captures the relation between the growth in the network's depth and the improvement in approximation.

We next go beyond pure expressivity analysis, and claim that the approximation curve plays a key role not only in approximation analysis, but also in predicting the success of gradient-based optimization algorithms. Specifically, we show that if the distribution is concentrated on the "fine" details of the fractal, then gradient-based optimization algorithms are likely to fail. In other words, the "stronger" the depth separation is (in the sense that shallow networks cannot even approximate the distribution) the harder it is to learn a deep network with a gradient-based algorithm. While we prove this statement for a specific fractal distribution, we state a conjecture aiming at formalizing this statement in a more general sense. Namely, we conjecture that a distribution which cannot be approximated by a shallow network cannot be learned using a gradient-based algorithm, even when using a deep architecture. We perform experiments on learning fractal distributions with deep networks trained with SGD and assert that the approximation curve has a crucial effect on whether a depth efficiency is observed or not. These results provide new insights as to when such deep distributions can be learned.

Admittedly, this paper is focused on analyzing a family of distributions that is synthetic by nature. That said, we note that the conclusions from this analysis may be interesting for the broader effort of understanding the power of depth in neural-networks. As mentioned, we show that there exist distributions with depth separation property that cannot be learned by gradient-based optimization algorithms. This result implies that any depth separation argument that does not consider the optimization process should be taken with a grain of salt. Additionally, our results hint that the success of learning deep networks depends on whether the distribution can be approximated by shallower networks. Indeed, this property is often observed in real-world distributions, where deeper networks perform better, but shallower networks exhibit good (if not perfect) performance. We demonstrate this behavior empirically on the CIFAR-10 dataset. The work of [1] shows similar behavior on the challenging ImageNet dataset, where a one-hidden layer network achieves 25% Top-5 accuracy (much better than a random guess), and a three-layer network already achieves 60%.

## 2   Preliminaries

Let $\mathcal{X} = \mathbb{R}^d$ be the domain space and $\mathcal{Y} = \{\pm 1\}$ be the label space. We consider distributions defined over sets generated by an iterated function system (IFS). An IFS is a method for constructing fractals, where a finite set of contraction mappings are applied iteratively, starting with some arbitrary initial set. Applying such process ad infinitum generates a self-similar fractal. In this work we will consider sets generated by performing a finite number of iterations from such process. We refer to the number of iterations of the IFS as the "depth" of the generated set.

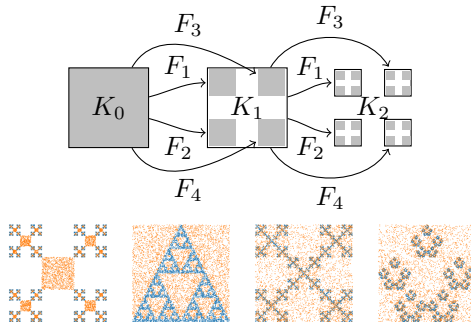

Figure 1: IFS and fractal distributions.

Formally, an IFS is defined by a set of $r$ contractive affine [1] transformations $F = (F_1, \ldots, F_r)$, where $F_i(\boldsymbol{x}) = \boldsymbol{M}^{(i)}\boldsymbol{x} + \boldsymbol{v}^{(i)}$ with full-rank matrix $\boldsymbol{M}^{(i)} \in \mathbb{R}^{d \times d}$, vector $\boldsymbol{v}^{(i)} \in \mathbb{R}^d$, s.t $\|F_i(\boldsymbol{x}) - F_i(\boldsymbol{y})\| < \|\boldsymbol{x} - \boldsymbol{y}\|$ for all $\boldsymbol{x}, \boldsymbol{y} \in \mathcal{X}$ (we use $\|\cdot\|$ to denote the $\ell_2$ norm, unless stated otherwise). We define the set $K_n \subseteq \mathcal{X}$ recursively by $K_0 = [-1, 1]^d$ and $K_n = F_1(K_{n-1}) \cup \cdots \cup F_r(K_{n-1})$. The IFS construction is shown in figure 1.

We define a "fractal distributions", denoted $\mathcal{D}_n$, to be any balanced distribution over $\mathcal{X} \times \mathcal{Y}$ such that positive examples are sampled from the set $K_n$ and negative examples are sampled from its complement. Formally, $\mathcal{D}_n = \frac{1}{2}(\mathcal{D}_n^+ + \mathcal{D}_n^-)$ where $\mathcal{D}_n^+$ is a distribution over $\mathcal{X} \times \mathcal{Y}$ that is supported on $K_n \times \{+1\}$, and $\mathcal{D}_n^-$ is a distribution over $\mathcal{X} \times \mathcal{Y}$ that is supported on $(\mathcal{X} \setminus K_n) \times \{-1\}$. Examples for such distributions are given in figure 1 and figure 2.

We consider the problem of learning fractal distributions with feed-forward ReLU neural-networks. A ReLU neural-network $\mathcal{N}_{\mathbf{W}, \boldsymbol{B}} : \mathcal{X} \to \mathcal{Y}$ of depth $t$ and width $k$ is defined recursively:

1. $\boldsymbol{x}^{(0)} = \boldsymbol{x}$

2. $\boldsymbol{x}^{(t')} = \sigma(\boldsymbol{W}^{(t')}\boldsymbol{x}^{(t'-1)} + \boldsymbol{b}^{(t')})$ for $1 \le t' \le t - 1$, and $\boldsymbol{W}^{(t')} \in \mathbb{R}^{k \times \dim \boldsymbol{x}^{(t'-1)}}, \boldsymbol{b}^{t'} \in \mathbb{R}^k$

3. $\mathcal{N}_{\mathbf{W}, \boldsymbol{B}}(\boldsymbol{x}) := \boldsymbol{x}^{(t)} = \boldsymbol{W}^{(t)}\boldsymbol{x}^{(t-1)} + \boldsymbol{b}^{(t)}$, for $\boldsymbol{W}^{(t)} \in \mathbb{R}^{1 \times k}, \boldsymbol{b}^{(t)} \in \mathbb{R}$

Where $\sigma(\boldsymbol{x}) := \max(\boldsymbol{x}, 0)$.

We denote by $\mathcal{H}_{k,t}$ the family of all functions that are implemented by a neural-network of width $k$ and depth $t$. Given a distribution $\mathcal{D}$ over $\mathcal{X} \times \mathcal{Y}$, we denote the error of a network $h \in \mathcal{H}_{k,t}$ on distribution $\mathcal{D}$ to be $L_\mathcal{D}(h) := \mathbb{P}_{(\boldsymbol{x},y) \sim \mathcal{D}_n} [\text{sign}(h(\boldsymbol{x})) \ne y]$. We denote the approximation error of $\mathcal{H}_{k,t}$ on $\mathcal{D}$ to be the minimal error of any such function: $L_\mathcal{D}(\mathcal{H}_{k,t}) := \min_{h \in \mathcal{H}_{k,t}} L_\mathcal{D}(h)$.

## 3 Expressivity and Approximation

In this section we analyze the expressive power of deep and shallow neural-networks w.r.t fractal distributions. We show two results. The first is a depth separation property of neural-networks. Namely, we show that shallow networks need an exponential number of neurons to realize such distributions, while deep networks need only a number of neurons that is linear in the problem's parameters. The second result bounds the approximation error achieved by networks that are not deep enough to achieve zero error. This bound depends on the specific properties of the fractal distribution.

We analyze IFSs where the images of the initial set $K_0$ under the different mappings do not overlap. This property allows the neural-network to "reverse" the process that generates the fractal structure. Additionally, we assume that the images of $K_0$ (and therefore the entire fractal), are contained in $K_0$, which means that the fractal does not grow in size. This is a technical requirement that could be achieved by correctly scaling the fractal at each step. While these requirements are not generally assumed in the context of IFSs, they hold for many common fractals. Formally, we assume:

**Assumption 1** *There exists $\epsilon > 0$ such that for $i \ne j \in [r]$ it holds that $d(F_i(K_0), F_j(K_0)) > \epsilon$, where $d(A, B) = \min_{\boldsymbol{x} \in A, \boldsymbol{y} \in B} \|\boldsymbol{x} - \boldsymbol{y}\|$.*

**Assumption 2** *For each $i \in [r]$ it holds that $F_i(K_0) \subseteq K_0$.*

As in many problems in machine learning, we assume the positive and negative examples are separated by some margin. Specifically, we assume that the positive examples are sampled from strictly inside the set $K_n$, with margin $\gamma$ from the set boundary. Formally, for some set $A$, we define $A^\gamma$ to be the set of all points that are far from the boundary of $A$ by at least $\gamma$: $A^\gamma := \{\boldsymbol{x} \in A : B_\gamma(\boldsymbol{x}) \subseteq A\}$, where $B_\gamma(\boldsymbol{x})$ denotes a ball around $\boldsymbol{x}$ with radius $\gamma$. So our assumption is the following:

**Assumption 3** *There exists $\gamma > 0$ such that $\mathcal{D}_n^+$ is supported on $K_n^\gamma \times \{+1\}$.*

We note that this assumption is used only in the proof of Theorem 1 below. In fact, a result similar to Theorem 1 can be shown without Assumption 3, as given in a recent preprint by [5].

## 3.1 Depth Separation

We show that neural-networks with depth linear in $n$ (where $n$ is the "depth" of the fractal) can achieve zero error on any fractal distribution satisfying the above assumptions, with only linear width. On the other hand, a shallow network needs a width exponential in $n$ to achieve zero error on such distributions. We start by giving the following expressivity result for deep networks:

**Theorem 1** *For any distribution $\mathcal{D}_n$ there exist neural-network of width $5dr$ and depth $2n + 1$, such that $L_{\mathcal{D}_n}(\mathcal{N}_{\mathbf{W},\mathbf{B}}) = 0$.*

We defer the proof of Theorem 1 to the appendix, and give here an intuition of how deep networks can express these seemingly complex distributions with a small number of parameters. Note that by definition, the set $K_n$ is composed of $r$ copies of the set $K_{n-1}$, mapped by different affine transformations. In our construction, each block of the network folds the different copies of $K_{n-1}$ on-top of each other, while "throwing away" the rest of the examples (by mapping them to a distinct value). The next block can then perform the same thing on all copies of $K_{n-1}$ together, instead of decomposing each subset separately. This allows a very efficient utilization of the network parameters.

The above result shows that deep networks require a number of parameters that grows linearly with $r$, $d$ and $n$ in order to realize any fractal distribution. Now, we want to consider the case of shallower networks, when the depth is not large enough to achieve zero error with linear width. Specifically, we show that when decreasing the depth of the network by a factor of $s$, we can achieve zero error by allowing the width to grow like $r^s$:

**Corollary 1** *For any distribution $\mathcal{D}_n$ and every natural $s \leq n$ there exists a neural-network of width $5dr^s$ and depth $2\lfloor n/s \rfloor + 2$, such that $L_{\mathcal{D}_n}(\mathcal{N}_{\mathbf{W},\mathbf{B}}) = 0$.*

This is an upper bound on the required width of a network that can realize $\mathcal{D}_n$, for any given depth. To show the depth separation property, we show that a shallow network needs an exponential number of neurons to realize any distribution without "holes" (areas of non-zero volume with no examples from $\mathcal{D}_n$, outside the margin area). This gives the equivalent lower bound on the required width:

**Theorem 2** *Let $\mathcal{D}_n$ be some fractal distribution, s.t for every ball $B \subseteq K_n^{\gamma} \cup (\mathcal{X} \setminus K_n)$ it holds that $\mathbb{P}_{(\boldsymbol{x},y) \sim \mathcal{D}_n}[\boldsymbol{x} \in B] > 0$. Then for every depth $t$ and width $k$, s.t $k < \frac{d}{e} r^{\frac{n}{td}}$, we have $L_{\mathcal{D}_n}(\mathcal{H}_{k,t}) > 0$.*

The previous result shows that in many cases we cannot guarantee exact realization of "deep" distributions by shallow networks that are not exponentially wide. On the other hand, we show that in some cases we may be able to give good guarantees on *approximating* such distributions with shallow networks, when we take into account how the examples are distributed within the fractal structure. We will formalize this notion in the next part of this section.

## 3.2 Approximation Curve

Given distribution $\mathcal{D}_n$, we define the **approximation curve** of this distribution to be the function $P : [n] \rightarrow [0, 1]$, where: $P(j) = \mathbb{P}_{(\boldsymbol{x},y) \sim \mathcal{D}_n}[\boldsymbol{x} \notin K_j \text{ or } y = 1]$. Notice that $P(0) = \frac{1}{2}$, $P(n) = 1$, and that $P$ is non-decreasing. The approximation curve $P$ captures exactly how the negative examples are distributed between the different levels of the fractal structure. If $P$ grows fast at the beginning, then the distribution is more concentrated on the low levels of the fractal (coarse details). If $P$ stays flat until the end, then most of the weight is on the high levels (fine details). Figure 2 shows samples from two distributions over the same fractal structure, with different approximation curves.

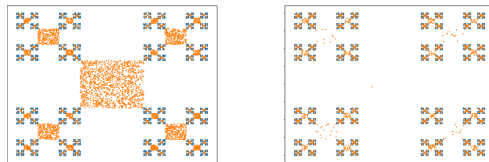

Figure 2: 2D cantor distributions of depth 5, negative examples in orange and positive in blue. The negative examples are concentrated in the middle rectangle, and not in all $\mathcal{X} \setminus K_n$. Left: "coarse" approximation curve (curve#1). Right: "fine" approximation curve (curve#4). The curves are as shown in figure 5 in the Experiments section.

A simple argument shows that distributions concentrated on coarse details can be well approximated by shallower networks. The following theorem characterizes the relation between the approximation curve and the "approximability" by networks of growing depth:

**Theorem 3** *Let $\mathcal{D}_n$ be some fractal distribution with approximation curve $P$. Fix some $j, s$, then for $\mathcal{H}_{k,t}$ with depth $t = 2\lfloor j/s \rfloor + 2$ and width $k = 5dr^s$, we have: $L_{\mathcal{D}_n}(\mathcal{H}_{k,t}) \leq 1 - P(j)$.*

This shows that using the approximation curve of distribution $\mathcal{D}_n$ allows us to give an upper bound on the approximation error for networks that are not deep enough. We give a lower bound for this error in a more restricted case. We limit ourselves to the case where $d = 1$, and observe networks of width $k < r^s$ for some $s$. Furthermore, we assume that the probability of seeing each subset of the fractal is the same. Then we get the following theorem:

**Theorem 4** *Assume that $\mathcal{D}_n$ is a distribution on $\mathbb{R}$ ($d = 1$). Note that for every $j$, $K_j$ is a union of $r^j$ intervals, and we denote $K_j = \cup_{i=1}^{r^j} I_i$ for intervals $I_i$. Assume that the distribution over each interval is equal, so for every $i, \ell, y'$: $\mathbb{P}_{(x,y) \sim \mathcal{D}_n}[x \in I_i \text{ and } y = y'] = \mathbb{P}_{(x,y) \sim \mathcal{D}_n}[x \in I_\ell \text{ and } y = y']$. Then for depth $t$ and width $k < r^s$, for $n > j > st$ we get: $L_{\mathcal{D}_n}(\mathcal{H}_{k,t}) \geq (1 - r^{st-j})(1 - P(j))$.*

The above theorem shows that for shallow networks, for which $st \ll j$, the approximation curve gives a very tight lower bound on the approximation error. This is due to the fact that shallow networks have a limited number of linear regions, and hence effectively give constant prediction on most of the "finer" details of the fractal distribution. This result implies that there are fractal distributions that are not only hard to realize by shallow networks, but that are even hard to approximate. Indeed, fix some small $\epsilon > 0$ and let $j := st + \log_r(\frac{1}{2\epsilon})$. Then if the approximation curve stays flat for the first $j$ levels (i.e $P(j) = \frac{1}{2}$), then from Theorem 4 the approximation error is at least $\frac{1}{2} - \epsilon$.

This gives a **strong depth separation** result: shallow networks have an error of $\approx \frac{1}{2}$ while a network of depth $t \geq 2\lfloor n/s \rfloor + 2$ can achieve zero error (on any fractal distribution). This strong depth separation result occurs when the distribution is concentrated on the "fine" details, i.e when the approximation curve stays flat throughout the "coarse" levels. In the next section we relate the approximation curve to the success of fitting a deep network to the fractal distribution, using gradient-based optimization algorithms. Specifically, we claim that distributions with strong depth separation **cannot** be learned by any network, deep or shallow, using gradient-based algorithms.

## 4 Optimization Analysis

So far, we analyzed the ability of neural-networks to express and approximate different fractal distributions. But it remains unclear whether these networks can be learned with gradient-based optimization algorithms. In this section, we show that the success of the optimization highly depends on the approximation curve of the fractal distribution. Namely, we show that for distributions with a "fine" approximation curve, that are concentrated on the "fine" details of the fractal, the optimization fails with high probability, for **any** gradient-based optimization algorithm.

To simplify the analysis, we focus in this section on a very simple fractal distribution: a distribution over the Cantor set in $\mathbb{R}$. The Cantor set $C_n$ is defined recursively by $C_0 = [0, 1]$ and $C_n = F_1(C_{n-1}) \cup F_2(C_{n-1})$, where $F_1(x) = \frac{1}{3} - \frac{1}{3}x$ and $F_2(x) = \frac{2}{3} + \frac{1}{3}x$. Now, fix margin $\gamma < \frac{3^{-n}}{2}$. We define the distribution $\mathcal{D}_n^+$ to be the uniform distribution over $C_n^\gamma \times \{+1\}$. The distribution $\mathcal{D}_n^-$ is a distribution over $C_0 \setminus C_n$, where we sample from each "level" $C_j$ ($j < n$) with probability $p_j$. Formally, we define $E_j := C_{j-1} \setminus C_j$ to be the $j$-th level of the negative distribution. We use $\mathcal{U}(E_j)$ to denote the uniform distribution on set $E_j$, then: $\mathcal{D}_n^- = \sum_{j=1}^{n} p_j (\mathcal{U}(E_j) \times \{-1\})$. Notice that the approximation curve of

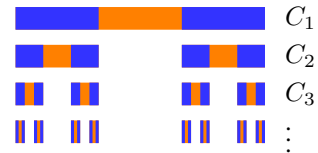

Figure 3: "Fine" Cantor distributions of growing depth. Negative areas in orange, positive in blue.

this distribution is given by: $P(j) = \frac{1}{2} + \frac{1}{2} \sum_{i=1}^{j} p_i$. As before, we wish to learn $\mathcal{D}_n = \frac{1}{2}(\mathcal{D}_n^+ + \mathcal{D}_n^-)$. Figure 3 shows a construction of such distribution.

The main theorem in this section shows the connection between the approximation curve and the behavior of a gradient-based optimization algorithm. This result shows that for deep enough Cantor

distributions, the value of the approximation curve on the fine details of the fractal bounds the norm of the population gradient for randomly initialized network:

**Theorem 5** *Fix some depth $t$, width $k$ and some $\delta \in (0,1)$. Let $n, n' \in \mathbb{N}$ such that $n > n' > \log^{-1}(\frac{3}{2})\log(\frac{4tk^2}{\delta})$. Let $\mathcal{D}_n$ be some Cantor distribution with approximation curve $P$. Assume we initialize a neural-network $\mathcal{N}_{\mathbf{W},\mathbf{B}}$ of depth $t$ and width $k$, with weights initialized uniformly in $[-\frac{1}{2n_{in}}, \frac{1}{2n_{in}}]$ (where $n_{in}$ denotes the in-degree of each neuron), and biases initialized with a fixed value $b = \frac{1}{2}$[2]. Denote the hinge-loss of the network on the population by: $\mathcal{L}(\mathcal{N}_{\mathbf{W},\mathbf{B}}) = \mathbb{E}_{(x,y)\sim\mathcal{D}^n}[\max\{1 - y\mathcal{N}_{\mathbf{W},\mathbf{B}}(x), 0\}]$. Then with probability at least $1 - \delta$ we have:*

1. $\left\| \frac{\partial}{\partial \mathbf{W}} \mathcal{L}(\mathcal{N}_{\mathbf{W},\mathbf{B}}) \right\|_{\max}, \left\| \frac{\partial}{\partial \mathbf{B}} \mathcal{L}(\mathcal{N}_{\mathbf{W},\mathbf{B}}) \right\|_{\max} \leq 5\left(P(n') - \frac{1}{2}\right)$

2. $L_{\mathcal{D}_n}(\mathcal{N}_{\mathbf{W},\mathbf{B}}) \geq \left(\frac{3}{2} - P(n')\right)(1 - P(n'))$

We now give some important implications of this theorem. First, notice that we can define Cantor distributions for which a gradient-based algorithm fails with high probability. Indeed, we define the "fine" Cantor distribution to be a distribution concentrated on the highest level of the Cantor set. Given our previous definition, this means $p_1, \ldots, p_{n-1} = 0$ and $p_n = 1$. The approximation curve for this distribution is therefore $P(0) = \cdots = P(n-1) = \frac{1}{2}$, $P(n) = 1$. Figure 3 shows the "fine" Cantor distribution drawn over its composing intervals. From Theorem 5 we get that for $n > \log^{-1}(\frac{3}{2})\log(\frac{4tk^2}{\delta})$, with probability at least $1 - \delta$, the population gradient is zero and the error is $\frac{1}{2}$. This result immediately implies that vanilla gradient-descent on the distribution will be stuck in the first step. But SGD, or GD on a finite sample, may move from the initial point, due to the stochasticity of the gradient estimation. What the theorem shows is that the objective is extremely flat almost everywhere in the regime of $\mathbf{W}$, so stochastic gradient steps are highly unlikely to converge to any solution with error better than $\frac{1}{2}$.

The above argument shows that there are fractal distributions that can be **realized** by deep networks, for which a standard optimization process is likely to fail. We note that this result is interesting by itself, in the broader context of depth separation results. It implies that for many deep architectures, there are distributions with depth separation property that cannot be learned by gradient-descent:

**Corollary 2** *There exist two constants $c_1, c_2$, such that for every width $k \geq 10$ and $\delta \in (0,1)$, for every depth $t > c_1 \log(\frac{k}{\delta}) + c_2$ there exists a distribution $\mathcal{D}$ on $\mathbb{R} \times \{\pm 1\}$ for which:*

1. $\mathcal{D}$ can be realized by a neural network of depth $t$ and width $10$.

2. $\mathcal{D}$ cannot be realized by a one-hidden layer network with less than $2^{t-1}$ units.

3. Any gradient-based algorithm trying to learn a neural-network of depth $t$ and width $k$, with initialization and loss described in Theorem 5, returns a network with error $\frac{1}{2}$ w.p $\geq 1 - \delta$.

We can go further, and use Theorem 5 to give a better characterization of these hard distributions. Recall that in the previous section we showed distributions that exhibit a *strong* depth separation property: distributions that are realizable by deep networks, for which shallow networks get an error exponentially close to $\frac{1}{2}$. From Theorem 5 we get that any Cantor distribution that gives a strong depth separation **cannot** be learned by gradient-based algorithms:

**Corollary 3** *Fix some depth $t$, width $k$ and some $\delta \in (0,1)$. Let $n > 4\log^{-1}(\frac{3}{2})\log(\frac{4tk^2}{\delta}) + 2$. Let $\mathcal{D}_n$ be some Cantor distribution such that any network of width $10$ and depth $t' < n$ has an error of at least $\frac{1}{2} - \epsilon^{n-t'}$, for some $\epsilon \in (0,1)$ (i.e, strong depth separation). Assume we initialize a network of depth $t$ and width $k$ as described in Theorem 5. Then with probability at least $1 - \delta$:*

1. $\left\| \frac{\partial}{\partial \mathbf{W}} \mathcal{L}(\mathcal{N}_{\mathbf{W},\mathbf{B}}) \right\|_{\max}, \left\| \frac{\partial}{\partial \mathbf{B}} \mathcal{L}(\mathcal{N}_{\mathbf{W},\mathbf{B}}) \right\|_{\max} \leq 5\epsilon^{n/2}$

2. $L_{\mathcal{D}_n}(\mathcal{N}_{\mathbf{W},\mathbf{B}}) \geq \frac{1}{2} - \frac{3}{2}\epsilon^{n/2}$

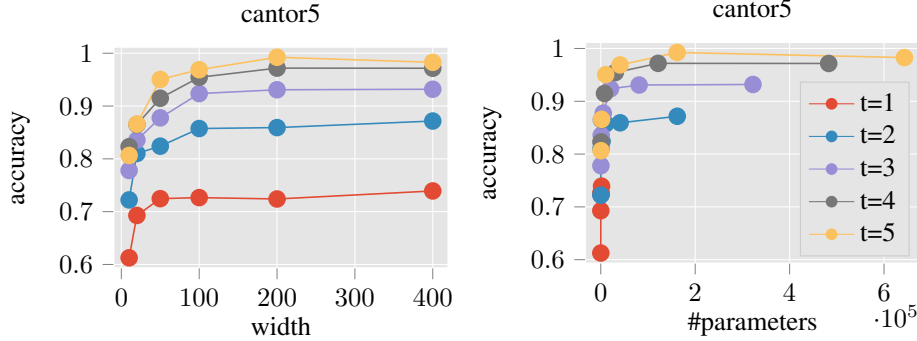

Figure 4: The effect of depth on learning the Cantor set.

This shows that in the strong depth separation case, the population gradient is exponentially close to zero with high probability. Effectively, this property means that even a small amount of stochastic noise in the gradient estimation (for example, in SGD), makes the algorithm fail.

This result gives a very important property of Cantor distributions. It shows that every Cantor distribution that cannot be **approximated** by a **shallow** network (achieving error greater than $\frac{1}{2}$), cannot be **learned** by a **deep** network (when training with gradient-based algorithms). While we show this in a very restricted case, we conjecture that this property holds in general:

**Conjecture 1** *Let $\mathcal{D}$ be some distribution such that $L_{\mathcal{D}}(\mathcal{H}_{k,t}) = 0$ (realizable with networks of width $k$ and depth $t$). If $L_{\mathcal{D}}(\mathcal{H}_{k,t'})$ is exponentially close to $\frac{1}{2}$ when $t' \to 1$, then any gradient-based algorithm training a network of depth $t$ and width $k$ will fail with high probability.*

We give an intuition of why such result may hold in a more general case. Note that a strong depth separation property means that the loss of any shallow network is close to a random guess, which implies that positive and negative examples are extremely hard to separate. In other words, any separation of the space to a small number of linear regions will have approximately the same amount of positive and negative examples in the same linear region. Our proof technique shows that in this case, upon initialization, a deep network only "rearranges" the linear regions, but does not change this property. Namely, when we initialize a deep network, most linear regions have a similar amount of positive and negative examples in them. In this case, gradient descent will fail, as the gradient will be approximately zero on all linear regions. We conjecture that a similar technique can be used in a more general setting. The use of fractal distributions greatly simplifies our analysis, but the core idea of why deep networks fail does not depend on the fractal structure of the distribution.

## 5 Experiments

In the previous section, we saw that learning a "fine" distribution with gradient-based algorithms is likely to fail. To complete the picture, we now assert that when the distribution has enough weight on the "coarse" details, SGD succeeds to learn a deep network with small error. Moreover, we show that when training on such distributions, a clear depth separation is observed, and deeper networks indeed perform better than shallow networks. Unfortunately, giving theoretical evidence to support this claim seems out of reach. Instead, we perform experiments to show these desired properties.

In this section we present our experimental results on learning deep networks with Adam optimizer ([7]). First, we show that depth separation is observed when training on samples from fractal distributions with "coarse" approximation curve: deeper networks perform better and have better parameter utilization. Second, we demonstrate the effect of training on "coarse" vs. "fine" distributions, showing that the performance of the network degrades as the approximation curve becomes finer. Finally, we analyze the behavior of networks of growing depth on CIFAR-10. We show that CIFAR-10 resembles the "coarse" fractal distribution in the sense that deep networks perform better, but shallow networks already give a good approximation.

We start by observing a distribution with a "coarse" approximation curve (denoted curve #1), where the negative examples are evenly distributed between the levels. The underlying fractal structure is a

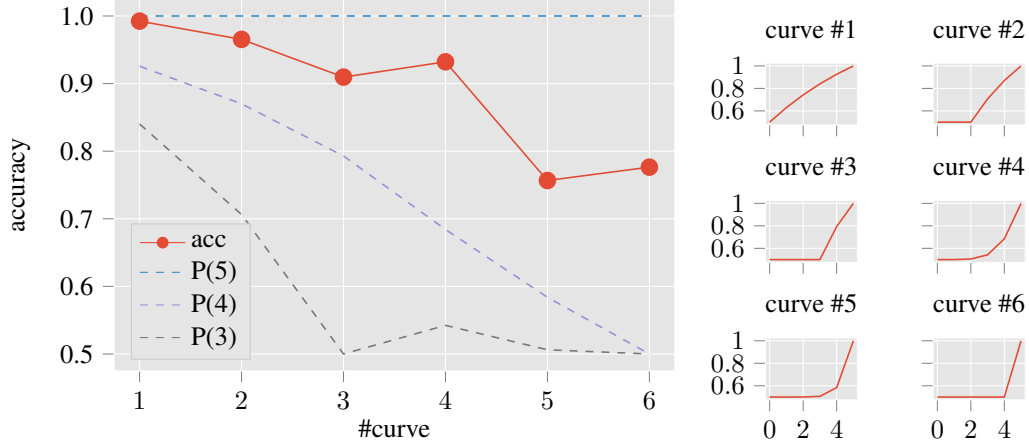

Figure 5: Learning depth 5 network on 2D Cantor set of depth 5, with different approximation curves. The figures show the values of the approximation curve (denoted $P$) at different levels of the fractal. Large values correspond to more weight. In red is the accuracy of the best depth 5 network architecture trained on these distributions.

two-dimensional variant of the Cantor set. This set is constructed by an IFS with four mappings, each one maps the structure to a rectangle in a different corner of the space. The negative examples are concentrated in the central rectangle of each structure. The distributions are shown in figure 2.

We train feed-forward networks of varying depth and width on a 2D Cantor distribution of depth 5. We sample 50K examples for a train dataset and 5K examples for a test dataset. We train the networks on this dataset with Adam optimizer for $10^6$ iterations, with batch size of $100$ and different learning rates. We observe the best performance of each configuration (depth and width) on the test data along the runs. The results of these experiments are shown in figure 4. In this experiment, we see that a wide enough depth 5 network gets almost zero error. Importantly, we can see a clear depth separation: deeper networks achieve better accuracy, and are more efficient in utilizing the network parameters.

Next, we observe the effect of the approximation curve on learning the distribution. We compare the performance of the best depth 5 networks, when trained on distributions with different approximation curves. The training and validation process is as described previously. We also plot the value of the approximation curve for each distribution, in levels $3, 4, 5$ of the fractal. The results of this experiment are shown in figure 5. Clearly, the approximation curve has a crucial effect on learning the distribution. While for "coarse" approximation curves the network achieves an error that is close to zero, distributions with "fine" approximation curves cause a drastic degradation in performance.

The degradation in performance becomes even more dramatic when considering deeper and more complex fractals. To demonstrate this, we ran an experiment on the Vicsek distribution of depth 6, where the examples are concentrated on the "fine" details of the fractal. Such distribution is hard to approximate by a shallow network, as shown in our theoretical analysis. We trained networks of various depth and width on this distribution, as described above. The results are shown in figure 6. As could be seen clearly, unlike distributions with "coarse" approximation curve (shown in figure 4), in this case the benefit of depth is not noticeable, and all architectures achieve an accuracy of slightly more than $0.5$ (i.e., chance level performance).

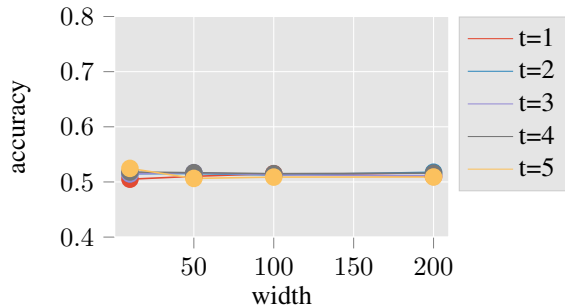

Figure 6: Performance on the "fine" Vicsek distribution, for networks of various depth and width.

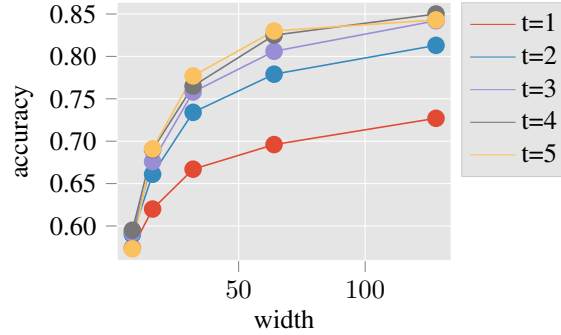

Figure 7: The effect of depth on learning CIFAR-10. We train CNNs with Adam for 60K steps. All layers are 5x5 Convolutions with ReLU activation, except the readout layer. We perform max-pool only in the first two layers. We use augmentations and training pipeline in [21].

We perform the same experiments with different fractal structures (figure 1 shows these distributions). Tables 1, 2 in the appendix summarize the results. We note that the effect of depth can be seen clearly in all fractal structures. The effect of the approximation curve is observed in all fractals, except the Sierpinsky Triangle (generated with 3 transformations), where the approximation curve seems to have no effect when the width of the network is large enough. This might be due to the fact that a depth 5 IFS with 3 transformations generates a small number of linear regions, making the problem overall relatively easy.

Finally, we want to show that the results given in this paper are interesting beyond the scope of our admittedly synthetic fractal distributions. We note that the use of fractal distributions is favorable from a theoretical perspective, as it allows us to develop crisp analysis and insightful results. On the other hand, it may raise a valid concern regarding the applicability of these results to real-world scenarios. To address this concern, we performed similar experiments on the CIFAR-10 data, studying the effect of width and depth on the performance of neural-networks on real data. The results are shown in figure 7. Notice that the trends on the CIFAR data resemble the behavior on the "coarse" fractal distributions. Importantly, note that the CIFAR data does not exhibit a *strong* depth separation, as depth gives only gradual improvement in performance. That is, while deeper networks indeed exhibit **better** performance, a shallow network already gives a **good** approximation. A similar behavior is observed even on the ImageNet dataset (see fig. 2 in [1]).

**Acknowledgements:** This research is supported by the European Research Council (TheoryDL project).

## Footnotes

[1] In general, IFSs can be constructed with non-linear transformations, but we discuss only affine IFS.

[2]We note that it is standard practice to initialize the bias to a fixed value. We fix $b = \frac{1}{2}$ for simplicity, but a similar result can be given for any choice of $b \in \left[0, \frac{1}{2}\right]$.

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
