[Supplementary Material]

## A Proof of Theorem 1 and Corollary 1

To prove the theorem, we begin with two technical lemmas:

**Lemma 1** *For every $\epsilon > 0$, there exists a neural-network of width $3d$ with two hidden-layers $(k = 3d, t = 3)$ such that $\mathcal{N}_{\mathbf{W},\mathbf{B}}(\mathbf{x}) = \mathbf{x}$ for $\mathbf{x} \in [0,1]^d$, and $\mathcal{N}_{\mathbf{W},\mathbf{B}}(\mathbf{x}) = 0$ for $\mathbf{x} \in \mathbb{R}^d$ with $d(\mathbf{x}, [0,1]^d) = \min_{\mathbf{y} \in [0,1]^d} \|\mathbf{x} - \mathbf{y}\| > \epsilon$.*

**Proof** Let $N > 0$ be some constant, and observe the function:

$$f_i(\mathbf{x}) = \sigma(\sigma(x_i) - N \sum_{j=1}^{d} \sigma(-x_j) - N \sum_{j=1}^{d} \sigma(x_j - 1))$$

Notice that $f_i(\mathbf{x}) = x_i$ for $\mathbf{x} \in [0,1]^d$, and that $f_i(\mathbf{x}) = 0$ if $d(\mathbf{x}, [0,1]^d) > \epsilon$, when taking $N$ to be large enough. Since $f(\mathbf{x}) = (f_1(\mathbf{x}), \ldots, f_d(\mathbf{x}))$ is a two hidden layer neural-network of width $3d$, the required follows. ∎

**Lemma 2** *For every $\gamma > 0$, there exists a neural-network of width $2d$ with two hidden-layers $(k = 2d, t = 3)$ such that $\mathcal{N}_{\mathbf{W},\mathbf{B}}(\mathbf{x}) = 1$ for $\mathbf{x} \notin [0,1]^d$, and $\mathcal{N}_{\mathbf{W},\mathbf{B}}(\mathbf{x}) = 0$ for $\mathbf{x} \in [\gamma, 1 - \gamma]^d$.*

**Proof** Let $N > 0$ be some constant, and observe the function:

$$\tilde{f}(\mathbf{x}) = 1 - \sigma(1 - N \sum_{j=1}^{d} \sigma(\gamma - x_j) - N \sum_{j=1}^{d} \sigma(x_j - 1 + \gamma))$$

Notice that $\tilde{f}(\mathbf{x}) = 0$ for $\mathbf{x} \in [\gamma, 1 - \gamma]^d$, and that $\tilde{f}(\mathbf{x}) = 0$ if $\mathbf{x} \notin [0,1]^d$, when taking $N$ to be large enough. Since $\tilde{f}$ a two hidden layer neural-network of width $2d$, the required follows. ∎

The next lemmas will show how a single block of the network operates on the set $K_n$:

**Lemma 3** *There exists a neural-network of width $\max\{dr, 3d\}$ with two hidden-layers $(k = 3dr, t = 3)$ such that for any $n$ we have:*

  *1. $\mathcal{N}_{\mathbf{W},\mathbf{B}}(K_n^\gamma) \subseteq K_{n-1}^\gamma$*

  *2. $\mathcal{N}_{\mathbf{W},\mathbf{B}}(K_1 \setminus K_n) \subseteq \mathcal{X} \setminus K_{n-1}$*

**Proof** As an immediate corollary from Lemma 1, there exists $f : \mathbb{R}^d \to \mathbb{R}^d$, that can be implemented by a neural network with two hidden-layers and width $3d$, such that $f(\mathbf{x}) = \mathbf{x}$ for $\mathbf{x} \in K_0$ and $f(\mathbf{x}) = 0$ if $d(\mathbf{x}, K_0) > \frac{\epsilon}{2}$. Define the following function:

$$g(\mathbf{x}) = \sum_{i=1}^{r} f\left((\mathbf{M}^{(i)})^{-1}\mathbf{x} - (\mathbf{M}^{(i)})^{-1}\mathbf{v}^{(i)}\right) = \sum_{i=1}^{r} f\left(F_i^{-1}(\mathbf{x})\right)$$

Notice that for every $\mathbf{x} \in \mathcal{X}$ there is at most one $i \in [r]$ such that $f(F_i^{-1}(\mathbf{x})) > 0$. Indeed, assume there are $i \neq j \in [r]$ such that $f(F_i^{-1}(\mathbf{x})) > 0$ and $f(F_j^{-1}(\mathbf{x})) > 0$. Therefore, $d(F_i^{-1}(\mathbf{x}), K_0) \leq \frac{\epsilon}{2}$ and $d(F_j^{-1}(\mathbf{x}), K_0) \leq \frac{\epsilon}{2}$. Therefore, there exist $\mathbf{y}, \mathbf{z} \in K_0$ such that $\left\|F_i^{-1}(\mathbf{x}) - \mathbf{y}\right\| \leq \frac{\epsilon}{2}$ and $\left\|F_j^{-1}(\mathbf{x}) - \mathbf{z}\right\| \leq \frac{\epsilon}{2}$. From this we get:

$$\|\mathbf{x} - F_i(\mathbf{y})\| \leq \left\|F_i^{-1}(\mathbf{x}) - \mathbf{y}\right\| \leq \frac{\epsilon}{2}$$

where we use the fact that $F_i$ is a contraction. Similarly, we get that $\|\mathbf{x} - F_j(\mathbf{z})\| \leq \frac{\epsilon}{2}$, so this gives us $\|F_i(\mathbf{y}) - F_j(\mathbf{z})\| \leq \epsilon$. Since $\mathbf{y}, \mathbf{z} \in K_0$, this is contradiction to Assumption 1.

We now show the following:

1. $g(K_n^\gamma) \subseteq K_{n-1}^\gamma$:

   Let $\boldsymbol{x} \in K_n^\gamma$, and denote $i \in [r]$ the unique $i$ for which $\boldsymbol{x} \in F_i(K_{n-1}) \subseteq F_i(K_0)$. From the properties of $f$, we get that $f(F_i^{-1}(\boldsymbol{x})) = F_i^{-1}(\boldsymbol{x})$ and $f(F_j^{-1}(\boldsymbol{x})) = 0$ for $j \neq i$, so $g(\boldsymbol{x}) = f(F_i^{-1}(\boldsymbol{x})) = F_i^{-1}(\boldsymbol{x}) \in K_{n-1}$. Since $\boldsymbol{x} \in K_n^\gamma$, we have $\boldsymbol{x} \in B_\gamma \subset K_n$ for some ball of radius $\gamma$ around $\boldsymbol{x}$. Note that since $F_i$ is a contraction, $F_i^{-1}$ is a linear expansive mapping, so $F_i^{-1}(B_\gamma)$ contains a ball of radius $\gamma$ around $F_i^{-1}(\boldsymbol{x})$, so $g(\boldsymbol{x}) \in K_{n-1}^\gamma$.

2. $g(K_1 \setminus K_n) \subseteq \mathcal{X} \setminus K_{n-1}$:

   Let $\boldsymbol{x} \in K_1 \setminus K_n$ and assume by contradiction that $g(\boldsymbol{x}) \in K_{n-1}$. Let $i \in [r]$ be the unique $i$ such that $\boldsymbol{x} \in F_i(K_0)$ and we have seen that in this case $g(\boldsymbol{x}) = F_i^{-1}(\boldsymbol{x})$, so $F_i^{-1}(\boldsymbol{x}) \in K_{n-1}$ and therefore $\boldsymbol{x} \in F_i(K_{n-1}) \subseteq K_n$ in contradiction to the assumption.

Since $g$ can be implemented with a neural network of width $3dr$ and two hidden-layer, this completes the proof of the lemma. ∎

**Lemma 4** *There exists a neural-network of width $2dr$ with two hidden-layers ($k = 2dr, t = 3$) such that for any $n$ we have:*

   *1. $\mathcal{N}_{\boldsymbol{W},\boldsymbol{B}}(\mathcal{X} \setminus K_1) = \{1\}$*

   *2. $\mathcal{N}_{\boldsymbol{W},\boldsymbol{B}}(K_1^\gamma) = \{0\}$*

**Proof** As a corollary of Lemma 2, there exists $\tilde{f} : \mathbb{R}^d \to \mathbb{R}$, a two hidden-layer neural-network of width $2d$, such that $\tilde{f}(\boldsymbol{x}) = 1$ for $\boldsymbol{x} \notin K_0$ and $\tilde{f}(\boldsymbol{x}) = 0$ for $\boldsymbol{x} \in K_0^\gamma$. Now, define:

$$\tilde{g}(\boldsymbol{x}) = 1 - r + \sum_{i=1}^{r} \tilde{f}\left(F_i^{-1}(\boldsymbol{x})\right)$$

We show the following:

1. $\tilde{g}(\mathcal{X} \setminus K_1) = \{1\}$:

   Let $\boldsymbol{x} \notin K_1 = \cup_i F_i(K_0)$, then for every $i$ we have $\boldsymbol{x} \notin F_i(K_0)$ and hence $F_i^{-1}(\boldsymbol{x}) \notin K_0$ so $\tilde{f}(F_i^{-1}(\boldsymbol{x})) = 1$ and so $\tilde{g}(\boldsymbol{x}) = 1$.

2. $\tilde{g}(K_1^\gamma) = \{0\}$:

   Let $\boldsymbol{x} \in K_1^\gamma$, and let $i$ be the unique index such that $\boldsymbol{x} \in F_i(K_0)$. So we have $\tilde{f}(F_i^{-1}(\boldsymbol{x})) = 0$ and for all $j \neq i$ we have $\tilde{f}(F_j^{-1}(\boldsymbol{x})) = 1$, and therefore $\tilde{g}(\boldsymbol{x}) = 0$.

And $\tilde{g}$ can be implemented by a width $2dr$ two hidden-layer network. ∎

**Proof** of Theorem 1 Let $g, \tilde{g}$ as defined in the previous lemmas. Denote $h_0 : \mathbb{R}^d \to \mathbb{R}^{d+1}$ the function:

$$h_0(\boldsymbol{x}) = [g(\boldsymbol{x}), \tilde{g}(\boldsymbol{x})]$$

and denote $h : \mathbb{R}^{d+1} \to \mathbb{R}^{d+1}$ the function:

$$h(\boldsymbol{x}) = [g(\boldsymbol{x}_{1\ldots d}), x_{d+1} + \tilde{g}(\boldsymbol{x}_{1\ldots d})]$$

Denote $h^n$ the composition of $h$ on itself $n$ times, and observe the network defined by $H = h^{n-1} \circ h_0$. Note that $H$ satisfies the following properties:

1. For $\boldsymbol{x} \in K_n^\gamma$ we have $H(\boldsymbol{x})_{d+1} = 0$: indeed, by iteratively applying the previous lemmas, we get that $g^j(\boldsymbol{x}) \in K_1^\gamma$ for every $j \leq n - 1$, and therefore $\tilde{g}(g^j(\boldsymbol{x})) = 0$ for every $j \leq n - 1$. Observe that: $H(\boldsymbol{x})_{d+1} = \sum_{j=1}^{n-1} \tilde{g}(g^j(\boldsymbol{x})) = 0$.

2. For $\boldsymbol{x} \notin K_n$ we have $H(\boldsymbol{x})_{d+1} \geq 1$: there exists $K_j$ such that $\boldsymbol{x} \in K_j \setminus K_{j+1}$, so by applying 3 we get $g^j(\boldsymbol{x}) \notin K_1$, so $\tilde{g}(g^j(\boldsymbol{x})) = 1$, and therefore $H(\boldsymbol{x})_{d+1} \geq 1$ (since the summation is over positive values).

470 Therefore, composing $H(\boldsymbol{x})$ with a linear threshold on $H(\boldsymbol{x})_{d+1}$ gives a network as required. Since
471 ever block of $H$ is has two hidden-layers of width $5dr$, we get that this network has depth $2n+1$ and
472 width $5dr$.

473 This shows that there exists a neural-network of width $5dr$ and depth $2n+1$, s.t $\text{sign}(\mathcal{N}_{\boldsymbol{W},\boldsymbol{B}}(K_n^\gamma)) = 1$
474 and $\text{sign}(\mathcal{N}_{\boldsymbol{W},\boldsymbol{B}}(\mathcal{X} \setminus K_n)) = -1$. Given the definition of the fractal distribution $\mathcal{D}_n$, this gives the
475 required. ∎

476

477 **Proof** of Corollary 1. Notice that for any $s$ that divides $n$, any IFS of depth $n$ with $r$ transformations
478 can be written as depth $\frac{n}{s}$ IFS with $r^s$ transformations. Indeed, for $\boldsymbol{i} = (i_1, \dots, i_s) \in [r]^s$ denote
479 $F_{\boldsymbol{i}}(\boldsymbol{x}) = F_{i_1} \circ \cdots \circ F_{i_s}(\boldsymbol{x})$, and we have: $K_s = \cup_{\boldsymbol{i} \in [r]^s} F_{\boldsymbol{i}}(K_0)$. So we can write a new IFS with
480 transformations $\{F_{\boldsymbol{i}}\}_{\boldsymbol{i} \in [r]^s}$, and these will generate $K_n$ in $\frac{n}{s}$ iterations. So, we can rewrite the IFS
481 with $r^s$ transformations, generating $K_{s \cdot \lfloor n/s \rfloor}$ in $\lfloor n/s \rfloor$ iterations. Therefore, using the construction
482 of Theorem 1, we have a network of depth $2\lfloor \frac{n}{s} \rfloor$ and width $5dr^s$ that maps $K_{\lfloor n/s \rfloor}$ to $K_0$, and
483 therefore maps $K_n$ to $K_{n-\lfloor n/s \rfloor}$. Now, a two hidden-layer network of width at most $dr^s$ can separate
484 $K_{n-\lfloor n/s \rfloor}^\gamma$ from $\mathcal{X} \setminus K_{n-\lfloor n/s \rfloor}$. This constructs a network of depth $2\lfloor n/s \rfloor + 2$ and width $5dr^s$ that
485 achieves the required. ∎

486

# B  Proof of Theorem 2

488 **Lemma 5** *Let $\mathcal{N}_{\boldsymbol{W},\boldsymbol{B}}$ be a network of depth $t$ and of width $k$, such that $\text{sign}(\mathcal{N}_{\boldsymbol{W},\boldsymbol{B}}(K_n^\gamma)) = 1$ and*
489 $\text{sign}(\mathcal{N}_{\boldsymbol{W},\boldsymbol{B}}(\mathcal{X} \setminus K_n)) = -1$. *Denote $s$ to be the ratio between the depth of the fractal and the depth*
490 *of the network, so $s := n/t$. Then the width of the network grows exponentially with $s$, namely:*
491 $k \geq \frac{d}{e} r^{s/d}$.

492 **Proof** From Proposition 3 in [8] we get that there are $\prod_{t'=1}^{t} \sum_{j=0}^{d} \binom{k}{j} \leq (ek/d)^{td}$ linear regions in
493 $\mathcal{N}_{\boldsymbol{W},\boldsymbol{B}}$ (where we use Lemma A.5 from [16]). Furthermore, every such linear region is an intersection
494 of affine half-spaces.

495 Note that any function such that $\text{sign}(f(K_n^\gamma)) = 1$ and $\text{sign}(f(\mathcal{X} \setminus K_n)) = -1$ has at least $r^n$ such
496 linear regions. Indeed, notice that $K_n = \cup_{\boldsymbol{i} \in [r]^n} F_{\boldsymbol{i}}(K_0)$. Assume by contradiction that there are
497 $< r^n$ linear regions, so there exists $\boldsymbol{i} \neq \boldsymbol{j} \in [r]^n$ such that $F_{\boldsymbol{i}}(K_0), F_{\boldsymbol{j}}(K_0)$ are in the same linear
498 region. Fix $\boldsymbol{x} \in F_{\boldsymbol{i}}(K_0)^\gamma, \boldsymbol{y} \in F_{\boldsymbol{j}}(K_0)^\gamma$ and observe the function $f$ along the line from $\boldsymbol{x}$ to $\boldsymbol{y}$. By
499 our assumption $f(\boldsymbol{x}) \geq 0, f(\boldsymbol{y}) \geq 0$. This line must cross $\mathcal{X} \setminus K_n$, since from Assumption 1 we get
500 that $d(F_{\boldsymbol{i}}(K_0), F_{\boldsymbol{j}}(K_0)) > 0$ for every $\boldsymbol{i} \neq \boldsymbol{j} \in [r]^n$. Therefore $f$ must get negative values along the
501 line between $\boldsymbol{x}$ to $\boldsymbol{y}$, so it must cross zero at least twice. Every linear region is an intersection of
502 half-spaces, and hence convex, so $f$ is linear on this path, and we reach a contradiction.

503 Therefore, we get that $(ek/d)^{td} \geq r^n$, and therefore: $k \geq \frac{d}{e} r^{s/d}$. ∎

504

505 **Proof** of the Theorem 2. Let $\mathcal{N}_{\boldsymbol{W},\boldsymbol{B}} \in \mathcal{H}_{k,t}$. From the above lemma, there exists $\boldsymbol{x} \in K_n^\gamma$ with
506 $\text{sign}(\mathcal{N}_{\boldsymbol{W},\boldsymbol{B}}(\boldsymbol{x})) = -1$ or otherwise there exists $\boldsymbol{x} \in \mathcal{X} \setminus K_n$ with $\text{sign}(\mathcal{N}_{\boldsymbol{W},\boldsymbol{B}}(\boldsymbol{x})) = 1$. Assume
507 w.l.o.g that we have $\boldsymbol{x} \in K_n^\gamma$ with $\text{sign}(\mathcal{N}_{\boldsymbol{W},\boldsymbol{B}}(\boldsymbol{x})) = -1$. Since $\mathcal{N}_{\boldsymbol{W},\boldsymbol{B}}$ is continuous, there exists
508 a ball around $\boldsymbol{x}$, with $\boldsymbol{x} \in B \subseteq K_n^\gamma$, such that $\text{sign}(\mathcal{N}_{\boldsymbol{W},\boldsymbol{B}}(B)) = -1$. From the properties of the
509 distribution we get:

$$\mathbb{P}_{(\boldsymbol{x},y) \sim \mathcal{D}_n} \left[ \text{sign}(\mathcal{N}_{\boldsymbol{W},\boldsymbol{B}}(\boldsymbol{x})) \neq y \right] \geq \mathbb{P}_{(\boldsymbol{x},y) \sim \mathcal{D}_n} \left[ \text{sign}(\mathcal{N}_{\boldsymbol{W},\boldsymbol{B}}(\boldsymbol{x})) \neq y \text{ and } \boldsymbol{x} \in B \right]$$
$$= \mathbb{P}_{(\boldsymbol{x},y) \sim \mathcal{D}_n} \left[ \boldsymbol{x} \in B \right] > 0$$

510 ∎
511

# C  Proof of of Theorem 3

513 **Proof** From Theorem 1 and Corollary 1, there exists a network of depth $t = 2\lfloor j/s \rfloor + 2$ and
514 width $5dr^s$ such that $\text{sign}(\mathcal{N}_{\boldsymbol{W},\boldsymbol{B}}(K_j^\gamma)) = 1$ and $\text{sign}(\mathcal{N}_{\boldsymbol{W},\boldsymbol{B}}(\mathcal{X} \setminus K_j)) = -1$. Notice that since

515   $K_n^\gamma \subseteq K_j^\gamma$, we have: $\mathbb{P}_{(\boldsymbol{x},y)\sim\mathcal{D}_n}\left[\boldsymbol{x} \notin K_j^\gamma \ and \ y = 1\right] = 0$. Therefore for this network we get:

516   $\mathbb{P}_{(\boldsymbol{x},y)\sim\mathcal{D}_n}\left[\mathrm{sign}(\mathcal{N}_{\mathbf{W},\boldsymbol{B}}(\boldsymbol{x})) \neq y\right] \leq \mathbb{P}_{(\boldsymbol{x},y)\sim\mathcal{D}_n}\left[x \in K_j \ and \ y \neq 1\right] = 1 - P(j).$ ∎

517

## D   Proof of Theorem 4

519 **Proof**  Using again [8], we get that the number of linear regions in $\mathcal{N}_{\mathbf{W},\boldsymbol{B}}$ is $r^{st}$. This means
520 that $\mathcal{N}_{\mathbf{W},\boldsymbol{B}}$ crosses zero at most $r^{st}$ times. Now, fix $n > j > st$, and notice that $K_j$ is a union
521 of $r^j$ intervals, so $K_j = \cup_{i=1}^{r^j} I_i$, for intervals $I_i$. By our assumption, we get that for every $i$,
522 $\mathbb{P}_{(x,y)\sim\mathcal{D}_n}\left[x \in I_i \ and \ y = -1\right] = p$ for some $p$, and from this we get:

$$\mathbb{P}_{(x,y)\sim\mathcal{D}_n}\left[x \in I_i \ and \ y = -1\right] = r^{-j}\mathbb{P}_{(x,y)\sim\mathcal{D}_n}\left[x \in K_j \ and \ y = -1\right]$$

523 We get that there are at most $r^{st}$ intervals of $K_j$ in which $\mathcal{N}_{\mathbf{W},\boldsymbol{B}}$ crosses zero. Denote $J \subseteq [r^j]$ the
524 subset of intervals on which $\mathrm{sign}(\mathcal{N}_{\mathbf{W},\boldsymbol{B}})$ is constant, and for every $i \in J$ we denote $\hat{y}_i$ such that
525 $\mathrm{sign}(\mathcal{N}_{\mathbf{W},\boldsymbol{B}}(I_i)) = \{\hat{y}_i\}$. Notice that:

$$\mathbb{P}_{(x,y)\sim\mathcal{D}_n}\left[x \in K_j \ and \ y = 1\right] = \frac{1}{2}$$
$$\mathbb{P}_{(x,y)\sim\mathcal{D}_n}\left[x \in K_j \ and \ y = -1\right] = 1 - P(j)$$

526 So the optimal choice for every $\hat{y}_i$ is 1. Then we have:

$$\begin{aligned}
\mathbb{P}_{(x,y)\sim\mathcal{D}_n}\left[\mathrm{sign}(\mathcal{N}_{\mathbf{W},\boldsymbol{B}}(x)) \neq y\right] &= \mathbb{P}_{(x,y)\sim\mathcal{D}_n}\left[\mathrm{sign}(\mathcal{N}_{\mathbf{W},\boldsymbol{B}}(x)) \neq -1 \ and \ x \notin K_j\right] \\
&+ \mathbb{P}_{(x,y)\sim\mathcal{D}_n}\left[\mathrm{sign}(\mathcal{N}_{\mathbf{W},\boldsymbol{B}}(x)) \neq y \ and \ x \in K_j\right] \\
&\geq \sum_{i\in[r^j]} \mathbb{P}_{(x,y)\sim\mathcal{D}_n}\left[\mathrm{sign}(\mathcal{N}_{\mathbf{W},\boldsymbol{B}}(x)) \neq y \ and \ x \in I_i\right] \\
&\geq \sum_{i\in J} \mathbb{P}_{(x,y)\sim\mathcal{D}_n}\left[\hat{y}_i \neq y \ and \ x \in I_i\right] \\
&\geq \sum_{i\in J} \mathbb{P}_{(x,y)\sim\mathcal{D}_n}\left[y = -1 \ and \ x \in I_i\right] \\
&= |J|r^{-j}\mathbb{P}_{(x,y)\sim\mathcal{D}_n}\left[x \in K_j \ and \ y = -1\right] \\
&\geq (1 - r^{st-j})(1 - P(j))
\end{aligned}$$

527 ∎
528

## E   Proof of Theorem 5

530 Observe that for every $n'$, we can write $C_{n'}$ as union of $2^{n'}$ intervals, so $C_{n'} = \cup_j I_j$. We can observe
531 the distribution limited to each of these intervals, and get the following:

532 **Lemma 6** *Let $\mathcal{D}_n$ be some cantor distribution (as defined in the paper). Then:*

$$\left|\mathbb{E}_{(x,y)\sim\mathcal{D}_n}\left[y\middle|x \in I_j\right]\right| \leq 2\left(P(n') - \frac{1}{2}\right)$$
$$\left|\mathbb{E}_{(x,y)\sim\mathcal{D}_n}\left[xy\middle|x \in I_j\right]\right| \leq 2\left(P(n') - \frac{1}{2}\right)$$

533 **Proof**  Let $I_j$ be some interval of $C_{n'}$, and let $c_j$ be the central point of $I_j$. Notice that by definition
534 of the distribution we have:

$$\mathbb{P}_{(x,y)\sim\mathcal{D}_n}\left[y = 1 \ and \ x \in I_j\right] = 2^{-n'-1}$$

$$\mathbb{P}_{(x,y)\sim\mathcal{D}_n}\left[y = -1 \ and \ x \in I_j\right] = 2^{-n'-1}(1 - \sum_{i=1}^{n'} p_i) = 2^{-n'}(1 - P(n'))$$

So we get that $\mathbb{P}_{(x,y)\sim\mathcal{D}_n}\left[x \in I_j\right] = 2^{-n'}(\frac{3}{2} - P(n'))$, and therefore:

$$\left|\mathbb{E}_{(x,y)\sim\mathcal{D}_n}\left[y\middle|x \in I_j\right]\right| = \left|\mathbb{P}_{(x,y)\sim\mathcal{D}_n}\left[y=1\middle|x \in I_j\right] - \mathbb{P}_{(x,y)\sim\mathcal{D}_n}\left[y=-1\middle|x \in I_j\right]\right|$$

$$= \left|(P(n') - \frac{1}{2})(\frac{3}{2} - P(n'))^{-1}\right|$$

$$\leq 2\left(P(n') - \frac{1}{2}\right)$$

Notice that from the structure of the set $C_n$, the average of all the points in $I_j \cap C_n$ is exactly the central point $c_j$ (this is due to the symmetry of the cantor set around its central point). Similarly, we get that each level of the negative distribution, $E_i := C_{i-1} \setminus C_i$, its average is also $c_j$. So we get:

$$\mathbb{E}_{(x,y)\sim\mathcal{D}_n}\left[x\middle|x \in I_j \text{ and } y=-1\right] = \mathbb{E}_{(x,y)\sim\mathcal{D}_n}\left[x\middle|x \in I_j \text{ and } y=1\right] = c_j$$

Therefore, we get that:

$$\mathbb{E}_{(x,y)\sim\mathcal{D}_n}\left[xy\middle|x \in I_j\right] = c_j\mathbb{P}_{(x,y)\sim\mathcal{D}_n}\left[y=1\middle|x \in I_j\right]$$

$$- c_j\mathbb{P}_{(x,y)\sim\mathcal{D}_n}\left[y=-1\middle|x \in I_j\right]$$

$$= c_j(P(n') - \frac{1}{2})(\frac{3}{2} - P(n'))^{-1}$$

So we have:

$$\left|\mathbb{E}_{(x,y)\sim\mathcal{D}_n}\left[xy\middle|x \in I_j\right]\right| \leq 2\left(P(n') - \frac{1}{2}\right)$$

∎

Using this result, we get the following lemma:

**Lemma 7** *Let $g : \mathbb{R} \to \mathbb{R}^k, f : \mathbb{R}^k \to \mathbb{R}$ two functions, and let $W \in \mathbb{R}^{k\times k}, c \in \mathbb{R}^k$, such that for every $j$, $g$ is affine on $I_j$ and $f$ is affine on $W g(I_j)+c \subseteq \mathbb{R}^k$. For every $j$, denote $u_j, v_j, a_j, b_j \in \mathbb{R}^k$ such that for every $x \in I_j$:*

$$g(x) = xu_j + a_j, \ f(Wg(x) + c) = v_j^\top(Wg(x) + c) + b_j$$

*Assume that $\|u_j\|_\infty, \|v_j\|_\infty, \|a_j\|_\infty, \|b_j\|_\infty \leq 1$. Denote $h : \mathbb{R} \to \mathbb{R}$ s.t $h(x) = f(Wg(x) + c)$. Then the following holds:*

$$\left\|\mathbb{E}_{(x,y)\sim\mathcal{D}_F}\left[-y\frac{\partial}{\partial W}h(x)\middle|x \in C_{n'}\right]\right\|_{\max} \leq 4\left(P(n') - \frac{1}{2}\right)$$

$$\left\|\mathbb{E}_{(x,y)\sim\mathcal{D}_F}\left[-y\frac{\partial}{\partial c}h(x)\middle|x \in C_{n'}\right]\right\|_\infty \leq 2\left(P(n') - \frac{1}{2}\right)$$

*Where for matrix $A$ we denote $\|A\|_{\max} = \max_{i,j}|a_{i,j}|$.*

**Proof** For every $x \in I_j$ it holds that:

$$\frac{\partial}{\partial W}h(x) = \frac{\partial}{\partial W}\left[v_j^\top(W(u_jx + a_j) + c) + b_j\right] = u_jv_j^\top x + a_jv_j^\top$$

$$\frac{\partial}{\partial c}h(x) = \frac{\partial}{\partial c}\left[v_j^\top(W(u_jx + a_j) + c) + b_j\right] = v_j$$

Using Lemma 6, we get:

$$\left\|\mathbb{E}_{(x,y)\sim\mathcal{D}_F}\left[-y\frac{\partial}{\partial\boldsymbol{W}}h(x)\Big|x\in I_j\right]\right\|_{\max} = \left\|\mathbb{E}_{(x,y)\sim\mathcal{D}_F}\left[-y(\boldsymbol{u}_j\boldsymbol{v}_j^\top x + \boldsymbol{a}_j\boldsymbol{v}_j^\top)\Big|x\in I_j\right]\right\|_{\max}$$
$$\leq \left|\mathbb{E}_{(x,y)\sim\mathcal{D}_F}\left[-yx\Big|x\in I_j\right]\right|\cdot\left\|\boldsymbol{u}_j\boldsymbol{v}_j^\top\right\|_{\max}$$
$$+ \left|\mathbb{E}_{(x,y)\sim\mathcal{D}_F}\left[-y\Big|x\in I_j\right]\right|\cdot\left\|\boldsymbol{a}_j\boldsymbol{v}_j^\top\right\|_{\max}$$
$$\leq 4\left(P(n') - \frac{1}{2}\right)$$

$$\left\|\mathbb{E}_{(x,y)\sim\mathcal{D}_F}\left[-y\frac{\partial}{\partial\boldsymbol{c}}h(x)\Big|x\in I_j\right]\right\|_{\infty} = \left\|\mathbb{E}_{(x,y)\sim\mathcal{D}_F}\left[-y\boldsymbol{v}_j\Big|x\in I_j\right]\right\|_{\infty}$$
$$= \left|\mathbb{E}_{(x,y)\sim\mathcal{D}_F}\left[-y\Big|x\in I_j\right]\right|\cdot\left\|\boldsymbol{v}_j\right\|_{\infty} \leq 2\left(P(n') - \frac{1}{2}\right)$$

Finally, from this we get:

$$\left\|\mathbb{E}_{(x,y)\sim\mathcal{D}_F}\left[-y\frac{\partial}{\partial\boldsymbol{W}}h(x)\Big|x\in C_{n'}\right]\right\|_{\max} \leq \sum_j \mathbb{P}\left[x\in I_j\right]\left\|\mathbb{E}_{(x,y)\sim\mathcal{D}_F}\left[-y\frac{\partial}{\partial\boldsymbol{W}}h(x)\Big|x\in I_j\right]\right\|_{\max}$$
$$\leq 4\left(P(n') - \frac{1}{2}\right)$$

$$\left\|\mathbb{E}_{(x,y)\sim\mathcal{D}_F}\left[-y\frac{\partial}{\partial\boldsymbol{c}}h(x)\Big|x\in C_{n'}\right]\right\|_{\infty} \leq \sum_j \mathbb{P}\left[x\in I_j\right]\left\|\mathbb{E}_{(x,y)\sim\mathcal{D}_F}\left[-y\frac{\partial}{\partial\boldsymbol{c}}h(x)\Big|x\in I_j\right]\right\|_{\infty}$$
$$\leq 2\left(P(n') - \frac{1}{2}\right)$$

∎

Now, we need to show that with high probability over the initialization of the network, every layer is affine on $I_j$-s.

**Lemma 8** *Fix $\delta \in (0,1)$, and let $s \leq k$. Let $g : \mathbb{R} \to \mathbb{R}^s$ such that for every $j$, $g$ is affine and non-expansive on $I_j$ (w.r.t to $\|\cdot\|_\infty$). Let $\boldsymbol{W} \in \mathbb{R}^{k\times s}$ a random matrix such that every entry is initialized uniformly from $[-\frac{1}{2s}, \frac{1}{2s}]$, and let $b > 2k^2\left(\frac{2}{3}\right)^{n'}\delta^{-1}$, some fixed bias. Denote $h(x) := \psi(\boldsymbol{W}g(x) + b)$, for some $\psi$ that is affine on every interval that is bounded away from zero. Then with probability at least $1 - \delta$, for every $j$, $h(x)$ is affine and non-expansive on $I_j$.*

**Proof** Denote $\boldsymbol{w}_i \in \mathbb{R}^k$ the $i$-th row of $\boldsymbol{W}$. Fix some $j$, and denote $c_j$ the central point of $I_j$. We show that:

$$\mathbb{P}_{\boldsymbol{w}_i\sim\mathcal{U}([-\frac{1}{2s},\frac{1}{2s}]^s)}\left[|\boldsymbol{w}_i^\top g(c_j) + b| \leq 3^{-n'}\right] \leq \frac{\delta}{k2^{n'}}$$

If $\|g(c_j)\|_\infty \leq b$ then $|\boldsymbol{w}_i^\top g(c_j)| \leq \|\boldsymbol{w}_i\|_1 \|g(c_j)\|_\infty \leq \frac{b}{2}$ and therefore $|\boldsymbol{w}_i^\top g(c_j) + b| \geq \frac{b}{2} > 3^{-n'}$. So we can assume $\|g(c_j)\|_\infty > b$, and let $\ell \in [k]$ be some index such that $g(c_j)_\ell > b$. Now, fix some values for $w_{i,1}, \ldots, w_{i,\ell-1}, w_{i,\ell+1}, \ldots, w_{i,k}$, and observe the distribution of $\boldsymbol{w}_i^\top g(c_j) + b$ (with respect to the randomness of $w_{i,\ell}$). Since $w_{i,\ell}$ is uniformly distributed in $[-\frac{1}{2s}, \frac{1}{2s}]$, we get that this

is a uniform distribution over some interval $J$ with $|J| \geq \frac{b}{s}$. From this we get:

$$\mathbb{P}_{w_{i,\ell} \sim \mathcal{U}([-\frac{1}{2s}, \frac{1}{2s}])} \left[ |\boldsymbol{w}_i^\top g(c_j) + b| \leq 3^{-n'} \right] = \mathbb{P}_{x \sim \mathcal{U}(J)} \left[ x \in [-3^{-n'}, 3^{-n'}] \right]$$
$$= |J \cap [-3^{-n'}, 3^{-n'}]|/|J|$$
$$\leq |[-3^{-n'}, 3^{-n'}]|/|J|$$
$$= \frac{2s3^{-n'}}{b} \leq \frac{\delta}{k2^{n'}}$$

Since there are $2^{n'}$ intervals $I_j$ and $k$ rows in $\boldsymbol{W}$, using the union bound we get that with probability at least $1 - \delta$, we have for all $j \in [2^{n'}]$ that $\|\boldsymbol{W}g(c_j) + b\|_\infty > 3^{-n'}$. Since we have $|I_j| = 3^{-n'}$, and since $g$ is non-expansive, this means that the set $\boldsymbol{W}g(I_j) + b$ does not cross zero at any of its coordinates. Indeed, assume there exists $i \in [k]$ such that $\boldsymbol{w}_i^\top g(I_j) + b$ crosses zero, and assume w.l.o.g that $\boldsymbol{w}_i^\top g(I_j) + b > 0$. Then there exists $x \in I_j$ with $\boldsymbol{w}_i^\top g(x) + b \leq 0$ and therefore:

$$3^{-n'} < |\boldsymbol{w}_i^\top g(x) - \boldsymbol{w}_i^\top g(c_j)| \leq \|\boldsymbol{w}_i\|_1 \|g(x) - g(c_j)\|_\infty \leq |x - c_j| \leq \frac{1}{2}3^{-n'}$$

and we reach contradiction.

Since $\psi$ is affine on intervals that are bounded away from zero, we get that $h(x) = \psi(\boldsymbol{W}g(x) + b)$ is affine on all $I_j$.

To show that $h$ is non-expansive on $I_j$, let $x, y \in I_j$, and from the fact that $g$ is non-expansive we have $\|g(x) - g(y)\|_\infty \leq |x - y|$. Since we showed that $\psi$ is affine on $\boldsymbol{W}g(I_j) + b$, we get:

$$|h(x) - h(y)| = |\psi(\boldsymbol{W}g(x) + b) - \psi(\boldsymbol{W}g(y) + b)| = |\psi(\boldsymbol{W}(g(x) - g(y)))| \leq |\boldsymbol{W}(g(x) - g(y))|$$

Therefore, for every $i$ we get:

$$|h(x)_i - h(y)_i| = |\boldsymbol{w}_i^\top (g(x) - g(y))| \leq \|\boldsymbol{w}_i\|_1 \|g(x)_j - g(y)_j\|_\infty \leq |x - y|$$

which completes the proof. ∎

**Lemma 9** *Let $g : \mathbb{R} \to \mathbb{R}^s$ such that $\|g(x)\|_\infty \leq 1$ for every $x \in [0,1]$. Let $\boldsymbol{W} \in \mathbb{R}^{k \times s}$ a random matrix such that every entry is initialized uniformly from $[-\frac{1}{2s}, \frac{1}{2s}]$, and let $0 < b \leq \frac{1}{2}$ some bias. Denote $h(x) := \sigma(\boldsymbol{W}g(x) + b)$. Then $\|h(x)\|_\infty \leq 1$ for every $x \in [0,1]$.*

**Proof** As before, we denote $\boldsymbol{w}_i$ the $i$-th row of $\boldsymbol{W}$, then for every $x \in [0,1]$ we get:

$$\|h(x)\|_\infty \leq \max_i |\boldsymbol{w}_i^\top g(x) + b| \leq \max_i \|\boldsymbol{w}_i\|_1 \|g(x)\|_\infty + b \leq 1$$

∎

Iteratively applying this lemma gives a bound on the norm of any hidden representation in the network:

**Lemma 10** *Assume we initialize a neural-network $\mathcal{N}_{\boldsymbol{W},\boldsymbol{B}}$ as described in Theorem 5, and denote $\mathcal{N}_{\boldsymbol{W},\boldsymbol{B}} = g^{(t)} \circ \cdots \circ g^{(1)}$. Denote $G^{(t')} = g^{(t')} \circ \cdots \circ g^{(1)}$, the output of the layer $t'$. Then for every layer $t'$ and for every $x \in [0,1]$ we get that: $\left\| G^{(t')}(x) \right\|_\infty \leq 1$.*

**Proof** First, $g^{(1)}(x) = \sigma(\boldsymbol{w}^{(1)}x + \boldsymbol{b}^{(1)})$, where $\boldsymbol{w}^{(1)} \sim \mathcal{U}([-\frac{1}{2}, \frac{1}{2}]^k)$ and $\boldsymbol{b}^{(1)} = [\frac{1}{2}, \ldots, \frac{1}{2}]$, so for every $x \in [0,1]$ we have:

$$\left\| g^{(1)}(x) \right\|_\infty \leq \left\| \boldsymbol{w}^{(1)} \right\|_\infty |x| + \frac{1}{2} \leq 1$$

Now, from Lemma 9, if $\left\| G^{(t')}(x) \right\|_\infty \leq 1$ for $x \in [0,1]$, then $\left\| G^{(t'+1)}(x) \right\|_\infty \leq 1$ for $x \in [0,1]$. By induction we get that $\left\| G^{(t')}(x) \right\|_\infty$ for $x \in [0,1]$ for every $t' \leq t - 1$. Finally, we have $\boldsymbol{w}^{(t)} \sim \mathcal{U}([-\frac{1}{2k}, \frac{1}{2k}])$ and $b^{(t)} = \frac{1}{2}$, and this gives us for every $x \in [0,1]$:

$$|\mathcal{N}_{\boldsymbol{W},\boldsymbol{B}}(x)| \leq |\boldsymbol{w}^{(t)} G^{(t-1)}(x) + b^{(t)}| \leq \left\| \boldsymbol{w}^{(t)} \right\|_1 \left\| G^{(t-1)}(x) \right\|_\infty + b^{(t)} \leq 1$$

■

602   We also show that the gradients are bounded on all the examples in the distribution:

603   **Lemma 11** *Assume we initialize a neural-network $\mathcal{N}_{\mathbf{W},\boldsymbol{B}}$ as described in Theorem 5. Then for every*
604   *layer $t'$, and every example $x \in [0,1]$ we have:*

$$\left\| \frac{\partial}{\partial \boldsymbol{W}^{(t')}} \mathcal{N}_{\mathbf{W},\boldsymbol{B}}(x) \right\|_{\max} \leq 1$$

$$\left\| \frac{\partial}{\partial \boldsymbol{b}^{(t')}} \mathcal{N}_{\mathbf{W},\boldsymbol{B}}(x) \right\|_{\infty} \leq 1$$

605   **Proof** Recall that we denote for every $x \in [0,1]$ the output of the layer $t'$ to be $x^{(t')} = G^{(t')}$. Denote
606   $\boldsymbol{D}^{(t')} = \operatorname{diag}(\sigma'(\boldsymbol{W}^{(t')}x^{(t')}))$. We calculate the gradient of the weights at layer $t'$:

$$\frac{\partial}{\partial \boldsymbol{W}^{(t')}} \mathcal{N}_{\mathbf{W},\boldsymbol{B}}(x) = \frac{\partial}{\partial \boldsymbol{W}^{(t')}} g^{(t)} \circ \cdots \circ g^{(t')}(\sigma(\boldsymbol{W}^{(t')}x^{(t'-1)} + \boldsymbol{b}^{(t')}))$$
$$= (\boldsymbol{w}^{(t)})^{\top} D^{(t-1)} \boldsymbol{W}^{(t-1)} \cdots D^{(t'+1)} \boldsymbol{W}^{(t'+1)} D^{(t')} x^{(t'-1)}$$

607   Denote $\|\cdot\|_{\infty}^{OP}$ the operator norm induced by $\ell_{\infty}$, and we get (using the properties of the weights
608   initialization):

$$\left\| D^{(t-1)} \boldsymbol{W}^{(t-1)} \cdots D^{(t'+1)} \boldsymbol{W}^{(t'+1)} D^{(t')} x^{(t'-1)} \right\|_{\infty} \leq \left\| D^{(t-1)} \right\|_{\infty}^{OP} \left\| \boldsymbol{W}^{(t-1)} \right\|_{\infty}^{OP} \cdots$$
$$\cdot \left\| D^{(t'+1)} \right\|_{\infty}^{OP} \left\| \boldsymbol{W}^{(t'+1)} \right\|_{\infty}^{OP} \left\| D^{(t')} \right\|_{\infty}^{OP} \left\| x^{(t'-1)} \right\|_{\infty}$$
$$\leq 1$$

609   And therefore: $\left\| \frac{\partial}{\partial \boldsymbol{W}^{(t')}} \mathcal{N}_{\mathbf{W},\boldsymbol{B}}(x) \right\|_{\max} \leq 1$ Finally, we calculate the gradient of the bias at layer $t'$:

$$\frac{\partial}{\partial \boldsymbol{b}^{(t')}} \mathcal{N}_{\mathbf{W},\boldsymbol{B}}(x) = \frac{\partial}{\partial \boldsymbol{b}^{(t')}} g^{(t)} \circ \cdots \circ g^{(t')}(\sigma(\boldsymbol{W}^{(t')}x^{(t'-1)} + \boldsymbol{b}^{(t')}))$$
$$= (\boldsymbol{w}^{(t)})^{\top} D^{(t-1)} \boldsymbol{W}^{(t-1)} \cdots D^{(t'+1)} \boldsymbol{W}^{(t'+1)} D^{(t')}$$

610   And since $\left\| \boldsymbol{w}^{(t)} \right\|_{\infty} \leq 1$ we get similarly to above that $\left\| \frac{\partial}{\partial \boldsymbol{b}^{(t')}} \mathcal{N}_{\mathbf{W},\boldsymbol{B}}(x) \right\|_{\infty} \leq 1$.   ■
611

612   **Proof** of Theorem 5. Denote each layer of the network by $g^{(i)}$, so we have: $\mathcal{N}_{\mathbf{W},\boldsymbol{B}}(x) = g^{(t)} \circ \cdots \circ$
613   $g^{(1)}(x)$. We show that two things hold on initialization:

614       1. $|\mathcal{N}_{\mathbf{W},\boldsymbol{B}}(x)| \leq 1$ for $x \in [0,1]$: immediately from Lemma 10.

615       2. With probability at least $1 - \delta$, for every $j$, $\mathcal{N}_{\mathbf{W},\boldsymbol{B}}$ is affine on $I_j$:

616           Denote $\hat{\delta} = \frac{\delta}{t}$, and notice that by the choice of $n'$, we get that $2k^2 \left(\frac{2}{3}\right)^{n'} \hat{\delta}^{-1} < \frac{1}{2} = b$.
617           Therefore, since $\sigma$ is affine on all intervals away from zero, we can apply Lemma 8 on
618           all the hidden layers of the network (choosing $s = 1, g = id$ for the first layer and
619           $s = k, g = g^{(t')} \circ g^{(1)}$ for the rest), and use union bound to get the required.

620   Now, to prove the theorem, observe that since $\mathcal{D}_n$ is supported on $[0,1] \times \{\pm 1\}$, we get that upon
621   initialization with probability 1 for $(x,y) \sim \mathcal{D}_n$ we have: $\max\{1 - y\mathcal{N}_{\mathbf{W},\boldsymbol{B}}(x), 0\} = 1 - y\mathcal{N}_{\mathbf{W},\boldsymbol{B}}(x)$.

622    Since $\mathcal{N}_{\mathbf{W},\boldsymbol{B}}(x)$ is affine on every $I_j$ w.p $1-\delta$, using Lemma 7 we get that in such case:

$$\left\|\frac{\partial}{\partial \mathbf{W}}\mathcal{L}(\mathcal{N}_{\mathbf{W},\boldsymbol{B}})\right\|_{\max} = \left\|\frac{\partial}{\partial \mathbf{W}}\mathbb{E}_{(x,y)\sim\mathcal{D}_n}\left[\max\{1-y\mathcal{N}_{\mathbf{W},\boldsymbol{B}}(x),0\}\right]\right\|_{\max}$$

$$= \left\|\mathbb{E}_{(x,y)\sim\mathcal{D}_n}\left[-y\frac{\partial}{\partial \mathbf{W}}\mathcal{N}_{\mathbf{W},\boldsymbol{B}}(x)\right]\right\|_{\max}$$

$$\leq \mathbb{P}_{(x,y)\sim\mathcal{D}_n}\left[x\in C_{n'}\right]\cdot\left\|\mathbb{E}_{(x,y)\sim\mathcal{D}_n}\left[-y\frac{\partial}{\partial \mathbf{W}}\mathcal{N}_{\mathbf{W},\boldsymbol{B}}(x)\Big|x\in C_{n'}\right]\right\|_{\max}$$

$$+\mathbb{P}_{(x,y)\sim\mathcal{D}_n}\left[x\notin C_{n'}\right]\cdot\left\|\mathbb{E}_{(x,y)\sim\mathcal{D}_n}\left[-y\frac{\partial}{\partial \mathbf{W}}\mathcal{N}_{\mathbf{W},\boldsymbol{B}}(x)\Big|x\notin C_{n'}\right]\right\|_{\max}$$

$$\leq 4\left(P(n')-\frac{1}{2}\right)+\left(P(n')-\frac{1}{2}\right)=5\left(P(n')-\frac{1}{2}\right)$$

623    and similarly we get $\left\|\frac{\partial}{\partial \boldsymbol{B}}\mathcal{L}(\mathcal{N}_{\mathbf{W},\boldsymbol{B}})\right\|_\infty \leq 3\left(P(n')-\frac{1}{2}\right)$.

624    To show that $\mathbb{P}_{(x,y)\sim\mathcal{D}_n}\left[\mathrm{sign}(\mathcal{N}_{\mathbf{W},\boldsymbol{B}}(x))\neq y\right]\geq \left(\frac{3}{2}-P(n')\right)(1-P(n'))$, observe that the sign
625    function is affine on intervals bounded away from zero. We can use Lemma 8 on the final layer, which
626    shows that $\mathrm{sign}(\mathcal{N}_{\mathbf{W},\boldsymbol{B}})$ is affine on the intervals $I_j$, so for every $I_j$ we get either $\mathrm{sign}(\mathcal{N}_{\mathbf{W},\boldsymbol{B}}(I_j))=$
627    $\{\hat{y}_j\}$ for some $\hat{y}_j\in\{\pm 1\}$. Now, using Lemma 7 we get that:

$$\mathbb{P}_{(x,y)\sim\mathcal{D}_n}\left[\mathrm{sign}(\mathcal{N}_{\mathbf{W},\boldsymbol{B}}(x))\neq y\Big|x\in I_j\right]=\mathbb{E}_{(x,y)\sim\mathcal{D}_n}\left[\frac{1}{2}-\frac{1}{2}y\hat{y}_j\Big|x\in I_j\right]$$

$$=\frac{1}{2}-\frac{1}{2}\hat{y}_j\mathbb{E}_{(x,y)\sim\mathcal{D}_n}\left[y\Big|x\in I_j\right]$$

$$\geq 1-P(n')$$

628    And from this we get:

$$\mathbb{P}_{(x,y)\sim\mathcal{D}_n}\left[\mathrm{sign}(\mathcal{N}_{\mathbf{W},\boldsymbol{B}}(x))\neq y\right]=\mathbb{P}_{(x,y)\sim\mathcal{D}_n}\left[x\in C_{n'}\right]\mathbb{P}_{(x,y)\sim\mathcal{D}_n}\left[\mathrm{sign}(\mathcal{N}_{\mathbf{W},\boldsymbol{B}}(x))\neq y\Big|x\in C_{n'}\right]$$

$$+\mathbb{P}_{(x,y)\sim\mathcal{D}_n}\left[x\notin C_{n'}\right]\mathbb{P}_{(x,y)\sim\mathcal{D}_n}\left[\mathrm{sign}(\mathcal{N}_{\mathbf{W},\boldsymbol{B}}(x))\neq y\Big|x\notin C_{n'}\right]$$

$$\geq \left(\frac{3}{2}-P(n')\right)(1-P(n'))$$

629
630

# F    Proof of Corollary 2 and Corollary 3

632    **Proof** of Corollary 2. Denote $a=2\log^{-1}(\frac{3}{2}), b=2\log^{-1}(\frac{3}{2})\log(\frac{4k^2}{\delta})+1$, and from Lemma
633    A.2 in [16], we get that if $t>4a\log(2a)+2b$ then $t>a\log(t)+b$. Choosing $n=\frac{t}{2}$ gives
634    $n>\log^{-1}(\frac{3}{2})\log(\frac{4tk^2}{\delta})+1$, so applying Theorem 5 shows that $\mathcal{D}_F^n$ satisfies 3. Theorem 1
635    immediately gives 1. Note that $\mathcal{D}_F^n$ can be realized only by functions with at least $2^{n-1}+1$ linear
636    regions. Shallow networks on $\mathbb{R}$ of width $k$ have at most $k+1$ linear regions, so this gives 2. ∎
637

638    **Proof** of Corollary 3. Using Theorem 3 and the strong depth separation property we get that
639    for every $t'$ we have: $1-P\left(\frac{t'-1}{2}\right)\geq L_{\mathcal{D}}(\mathcal{H}_{10,t'})\geq \frac{1}{2}-\epsilon^{n-t'}$. Choosing $t'=\frac{n}{2}$ and taking
640    $n'=\frac{n}{4}-\frac{1}{2}$ we get $P(n')\leq \frac{1}{2}+\epsilon^{n/2}$. By the choice of $n$ we can apply Theorem 5 and get the
641    required. ∎
642

# G   Experimental Results

The following tables summarize the results of all the experiments that are detailed in Section 5.
Table 1: Performance of different network architectures on various fractal distributions of depth 5,
with different fractal structures.

| DEPTH / WIDTH | 10 | 20 | 50 | 100 | 200 | 400 |
|---|---|---|---|---|---|---|
| SIERPINSKY TRIANGLE | | | | | | |
| 1 | 0.78 | 0.82 | 0.88 | 0.87 | 0.89 | 0.90 |
| 2 | 0.86 | 0.91 | 0.93 | 0.94 | 0.95 | 0.95 |
| 3 | 0.89 | 0.92 | 0.96 | 0.96 | 0.97 | 0.97 |
| 4 | 0.87 | 0.94 | 0.97 | 0.97 | 0.97 | 0.98 |
| 5 | 0.89 | 0.94 | 0.96 | 0.97 | 0.97 | 0.98 |
| 2D CANTOR SET | | | | | | |
| 1 | 0.61 | 0.69 | 0.72 | 0.73 | 0.72 | 0.74 |
| 2 | 0.72 | 0.81 | 0.82 | 0.86 | 0.86 | 0.87 |
| 3 | 0.78 | 0.84 | 0.88 | 0.92 | 0.93 | 0.93 |
| 4 | 0.82 | 0.86 | 0.91 | 0.95 | 0.97 | 0.97 |
| 5 | 0.81 | 0.87 | 0.95 | 0.97 | 0.99 | 0.98 |
| PENTAFLAKE | | | | | | |
| 1 | 0.66 | 0.65 | 0.67 | 0.70 | 0.76 | 0.76 |
| 2 | 0.71 | 0.73 | 0.79 | 0.81 | 0.82 | 0.83 |
| 3 | 0.73 | 0.78 | 0.83 | 0.84 | 0.85 | 0.86 |
| 4 | 0.76 | 0.79 | 0.85 | 0.87 | 0.88 | 0.88 |
| 5 | 0.76 | 0.81 | 0.86 | 0.88 | 0.87 | 0.90 |
| VICSEK | | | | | | |
| 1 | 0.59 | 0.60 | 0.63 | 0.66 | 0.67 | 0.68 |
| 2 | 0.64 | 0.70 | 0.72 | 0.75 | 0.76 | 0.75 |
| 3 | 0.69 | 0.72 | 0.77 | 0.79 | 0.81 | 0.82 |
| 4 | 0.71 | 0.74 | 0.79 | 0.82 | 0.83 | 0.84 |
| 5 | 0.70 | 0.77 | 0.82 | 0.84 | 0.86 | 0.86 |

Table 2: Performance of depth 5 network on the different fractal structure (of depth 5), with varying approximation curves.

| Curve # / Width | 10 | 20 | 50 | 100 | 200 | 400 |
|---|---|---|---|---|---|---|
| SIERPINSKY TRIANGLE | | | | | | |
| 1 | 0.89 | 0.94 | 0.96 | 0.97 | 0.97 | 0.98 |
| 2 | 0.89 | 0.94 | 0.96 | 0.97 | 0.97 | 0.97 |
| 3 | 0.78 | 0.94 | 0.96 | 0.97 | 0.97 | 0.97 |
| 4 | 0.79 | 0.92 | 0.96 | 0.96 | 0.97 | 0.97 |
| 5 | 0.76 | 0.89 | 0.96 | 0.97 | 0.97 | 0.97 |
| 6 | 0.76 | 0.90 | 0.97 | 0.97 | 0.98 | 0.98 |
| 2D CANTOR SET | | | | | | |
| 1 | 0.81 | 0.87 | 0.95 | 0.97 | 0.99 | 0.98 |
| 2 | 0.70 | 0.85 | 0.92 | 0.94 | 0.94 | 0.97 |
| 3 | 0.62 | 0.73 | 0.75 | 0.80 | 0.91 | 0.89 |
| 4 | 0.53 | 0.65 | 0.77 | 0.77 | 0.84 | 0.93 |
| 5 | 0.57 | 0.61 | 0.65 | 0.69 | 0.76 | 0.73 |
| 6 | 0.53 | 0.64 | 0.66 | 0.78 | 0.71 | 0.61 |
| PENTAFLAKE | | | | | | |
| 1 | 0.76 | 0.81 | 0.86 | 0.88 | 0.87 | 0.90 |
| 2 | 0.59 | 0.68 | 0.77 | 0.78 | 0.80 | 0.84 |
| 3 | 0.54 | 0.57 | 0.64 | 0.63 | 0.72 | 0.64 |
| 4 | 0.53 | 0.55 | 0.58 | 0.61 | 0.65 | 0.68 |
| 5 | 0.52 | 0.52 | 0.52 | 0.55 | 0.60 | 0.57 |
| 6 | 0.52 | 0.52 | 0.52 | 0.53 | 0.56 | 0.54 |
| VICSEK | | | | | | |
| 1 | 0.70 | 0.77 | 0.82 | 0.84 | 0.86 | 0.86 |
| 2 | 0.59 | 0.61 | 0.67 | 0.69 | 0.71 | 0.71 |
| 3 | 0.56 | 0.55 | 0.58 | 0.64 | 0.64 | 0.65 |
| 4 | 0.51 | 0.52 | 0.54 | 0.56 | 0.58 | 0.59 |
| 5 | 0.52 | 0.52 | 0.52 | 0.55 | 0.53 | 0.57 |
| 6 | 0.53 | 0.51 | 0.51 | 0.55 | 0.58 | 0.59 |