[Reviews · NeurIPS 2019]

Reviewer 1



This is a good paper that suggests excellent directions for new work. The key point is captured in this statement: "we conjecture that a distribution which cannot be approximated by a shallow network cannot be learned using a gradient-based algorithm, even when using a deep architecture." The authors provide first steps towards investigating this claim. There has been a small amount of work on the typical expressivity of neural networks, in addition to the "worst-case approach." See the papers "Complexity of linear regions in deep networks" and "Deep ReLU Networks Have Surprisingly Few Activation Patterns" by Hanin and Rolnick, which prove that while the number of linear regions can be made to grow exponentially with the depth, the typical number of linear regions is much smaller. See also "Do deep nets really need to be deep?" by Ba and Caruana, which indicates that once deep networks have learned a function, shallow networks can often be trained to distill the deep networks without appreciable performance loss. On this note, the authors write: "Many of these works consider various measures of “complexity” that grow exponentially fast with the depth of the network, but not with the width." As stated, this is not quite right - these works describe measures of complexity that *can* grow exponentially fast with the depth. (Note also that for this reason, line 519 in the appendix should read "the number of linear regions...is *at most* r^{st}" - though this does not seem to affect the proof.) Figure 2 doesn't actually show any approximation curves, though it claims to do so. Rather, it shows two different distributions associated with the same fractal. It would be instructive to see plots of these approximation curves. In Figure 5, the main plot can be made clearer, indicating exactly what the curves are that are shown. As a minor point, "Cantor" should always be capitalized. Line 30: "analyzing" should be "analyze."

Reviewer 2



This paper is about expressive abilities of deep neural networks and the related gradient-based optimization processes for distributions generated by iterated function systems. The first result shows that such a distribution can be expressed efficiently by deep neural networks, but not by shallow ones. The second result shows that gradient-based algorithms are likely fail in processing such a distribution. A key assumption is Assumption 3 requiring that the positive class is contained in the set of points that are far away from the boundary of the n-th iterated set K_n by at least gamma. Note that the limiting set of K_n is a fractal set having no interior point in most cases. So the required margin gamma depends on n and would be extremely small when n is large. This observation leads to the reviewer's concerns over the main results: the nice depth separation argument is too special, and the evidence for the failure of gradient-based algorithms is not strong enough.

Reviewer 3



Originality: This work shows that a family of distributions (fractal distribution) can be used for having a separation argument between shallow and deep networks. Moreover, based on a notion of the approximation curve, we can study the dynamics of training deep neural networks. Quality: It seems that the submission is technically sound but I did not go through the proofs. Clarity: The submission is well written. Significance: The authors proposed an interesting depth separation argument based on fractal distributions. More importantly, they related this argument to the success of training deep neural networks by gradient-based algorithms. =================== Post-rebuttal comments: =================== Thanks for providing new experimental result. It might be good to add this result to the paper to support your conjecture.

[Author Response · NeurIPS 2019]

We thank the reviewers for their overall positive feedback.

The main concern raised by **Reviewer #2** is that Assumption 3, which requires that the positive examples are sampled
within a margin $\gamma$ from the boundaries of the set $K_n$, makes the overall results too weak. Reviewer #2 rightly noted
that for the distributions presented in the paper, $\gamma$ should decay exponentially with $n$, and this may seem to be a strong
requirement. While this is a valid concern, we stress that we did not use this assumption at all for the results on failure
of gradient-descent. Since having a margin $\gamma$ can only help the optimization, dropping this assumption simply makes
the optimization harder, hence these results still hold. Similarly, other negative results in the paper, namely - the
inability of shallow networks to express fractal distributions, hold without Assumption 3: this assumption only makes
the approximation problem easier.

In fact, the only results that rely on Assumption 3 is Theorem 1 and its corollaries, which give positive results, stating
that fractal distributions can be efficiently expressed by deep networks. While the existence of a margin simplifies
the construction made in the proof of this theorem, we can prove this theorem even **without** Assumption 3. We give
a sketch of such proof below. **To summarize, in order to answer the concern of Reveiwer #2 we will completely**
**remove Assumption 3 from the final version, and adjust all the theorems accordingly.**

Following the suggestion of **Reviewer #3**, we ran an ex-
periment on the Vicsek distribution of depth 6, where
the examples are concentrated on the "fine" details of
the fractal. Such distribution is hard to approximate by
a shallow network, as shown in our theoretical analysis.
We trained networks of various depth and width on this
distribution (as in the experiments described in the orig-
inal submission). The results are shown in Figure 1. As
could be seen clearly, unlike distributions with "coarse"
approximation curve (shown in the original submission),
in this case the benefit of depth is not noticeable, and all
architectures achieve an accuracy of slightly more than
0.5 (i.e., chance level performance).

We will additionally fix other minor issues raised by the
reviewers in the final version.

Figure 1: Performance on the "fine" Vicsek distribution.

**Proof Sketch of Theorem 1 without Assumption 3**

**Lemma 2** *(without Assumption 3, standard construction of a ReLU network) There exists a neural-network with two*
*hidden-layers such that $\mathcal{N}_{\mathbf{W},\mathbf{B}}(\boldsymbol{x}) < 0$ for $\boldsymbol{x} \notin [0,1]^d$, and $\mathcal{N}_{\mathbf{W},\mathbf{B}}(\boldsymbol{x}) \geq 0$ for $\boldsymbol{x} \in [0,1]^d$.*

**Lemma 3** *(without Assumption 3) There exists a neural-network of width $\max\{dr, 3d\}$ with two hidden-layers ($k =$*
*$3dr, t = 3$) such that for any $n$ we have: $\mathcal{N}_{\mathbf{W},\mathbf{B}}(K_n) \subseteq K_{n-1}$ and $\mathcal{N}_{\mathbf{W},\mathbf{B}}(K_1 \setminus K_n) \subseteq \mathcal{X} \setminus K_{n-1}$*

**Proof** Simple modification to the proof of Lemma 3 in the original submission. ∎
36

**Lemma 4** *(without Assumption 3) There exists a neural-network of width $2dr$ with two hidden-layers ($k = 2dr, t = 3$)*
*such that for any $n$ we have: $\mathcal{N}_{\mathbf{W},\mathbf{B}}(\mathcal{X} \setminus K_1) < 0$ and $\mathcal{N}_{\mathbf{W},\mathbf{B}}(K_1) \geq 0$.*

**Proof** Using Lemma 2, and following the same proof of Lemma 4 in the original submission. ∎
40

**Proof** of Theorem 1 (without Assumption 3). We follow a proof similar to the one given in the original submission.
Instead of the original definition of $h$, we define $h(\boldsymbol{x}) = [g(\boldsymbol{x}_{1...d}), \boldsymbol{x}_{d+1} - \sigma(\boldsymbol{x}_{d+1} - \tilde{g}(\boldsymbol{x}_{1...d}))]$. Then, constructing
$H$ as in the original proof satisfies that $H(\boldsymbol{x})_{d+1} < 0$ if and only if $x \notin K_n$: if the $d+1$ coordinate of some layer
becomes negative, it stays negative throughout the network (since the $d+1$ coordinate of each layer is just the minimum
of the $d+1$ coordinates of previous layers). Therefore, a network that outputs $H(\boldsymbol{x})_{d+1}$ achieves the required. ∎
46

[Meta-Review · NeurIPS 2019]

This paper investigates the effect of depth of expressivity and learnability, given a distribution generated by an iterated function system. In particular, they showed that shallow networks need an exponential number of neurons to realize a fractal distribution while deep networks only require a number of neurons that is linear with the depth of the fractal distribution. The results are interesting and could shed some lights on the theoretical understanding of deep learning. So, the reviewers have shown their support to this paper, despite that it studies a mathematically narrow case whose practical value is not very clear. The impact of the work will be greatly improved if the authors could extend their studies to more general cases.